# Characterizing possible failure modes in physics-informed neural networks

**Aditi S. Krishnapriyan**[*,1,2], **Amir Gholami**[*,2],
**Shandian Zhe**[3], **Robert M. Kirby**[3], **Michael W. Mahoney**[2,4]
[1]Lawrence Berkeley National Laboratory, [2]University of California, Berkeley,
[3]University of Utah, [4]International Computer Science Institute
{aditik1, amirgh, mahoneymw}@berkeley.edu, {zhe, kirby}@cs.utah.edu

## Abstract

Recent work in scientific machine learning has developed so-called physics-informed neural network (PINN) models. The typical approach is to incorporate physical domain knowledge as soft constraints on an empirical loss function and use existing machine learning methodologies to train the model. We demonstrate that, while existing PINN methodologies can learn good models for relatively trivial problems, they can easily fail to learn relevant physical phenomena for even slightly more complex problems. In particular, we analyze several distinct situations of widespread physical interest, including learning differential equations with convection, reaction, and diffusion operators. We provide evidence that the soft regularization in PINNs, which involves PDE-based differential operators, can introduce a number of subtle problems, including making the problem more ill-conditioned. Importantly, we show that these possible failure modes are not due to the lack of expressivity in the NN architecture, but that the PINN's setup makes the loss landscape very hard to optimize. We then describe two promising solutions to address these failure modes. The first approach is to use curriculum regularization, where the PINN's loss term starts from a simple PDE regularization, and becomes progressively more complex as the NN gets trained. The second approach is to pose the problem as a sequence-to-sequence learning task, rather than learning to predict the entire space-time at once. Extensive testing shows that we can achieve up to 1-2 orders of magnitude lower error with these methods as compared to regular PINN training.

## 1  Introduction

Partial differential equations (PDEs) are commonly used to describe different phenomena in science and engineering. These PDEs are often derived by starting from governing first principles (e.g., conservation of mass or energy). It is typically not possible to find analytical solutions to these PDEs for many real-world settings. Thus, many different numerical methods (e.g., the finite element method [44], pseudo-spectral methods [9], etc.) have been introduced to approximate their solutions/behavior. However, these PDEs can be quite complex for several settings (e.g., turbulence simulations), and numerical integration techniques, which typically update and improve a candidate solution iteratively until convergence, are often quite computationally expensive. Motivated by this—as well as the increasing quantities of data available in many scientific and engineering applications—there has been recent interest in developing machine learning (ML) approaches to find the solution of the underlying PDEs (and/or work in tandem with numerical solutions). As a result, the area of Scientific Machine Learning (SciML)—which aims to couple traditional scientific mechanistic modeling

---

[*]Equal contribution

35th Conference on Neural Information Processing Systems (NeurIPS 2021).

(typically, differential equations) with data-driven ML methodologies (most recently, neural network training)—has emerged. In this vein, there have been a number of ML approaches to incorporate scientific knowledge into such problems while keeping the automatic, data-driven estimates of the solution [2, 17, 33, 39].

A recent line of work involves Physics-Informed Neural Network (PINN) models, which aim to incorporate physical domain knowledge as soft constraints on an empirical loss function, that is then optimized using existing ML training methodologies. To some degree, PINNs are an example of "grafting together" domain-driven models and data-driven methodologies. However, there are important subtleties with this, and we identify several possible failure modes with a naive approach. We then illustrate possible directions for addressing these failure modes.

**Background and problem overview.**    Many of the problems with a PDE constraint fit the following abstraction:
$$\mathcal{F}(u(x,t)) = 0, \qquad x \in \Omega \subset \mathbb{R}^d, \quad t \in [0,T], \tag{1}$$
where $\mathcal{F}$ is a differential operator representing the PDE, $u(x,t)$ is the state variable (i.e., parameter of interest), $x/t$ denote space/time, $T$ is the time horizon, and $\Omega$ is the spatial domain. Since $\mathcal{F}$ is a differential operator, in general one must specify appropriate boundary and/or initial conditions to ensure the existence/uniqueness of a solution to Eq. 1. In the context of PDEs, $\mathcal{F}$ can be taxonomized into a parabolic, hyperbolic, or elliptic differential operator [23]. Quintessential examples of $\mathcal{F}$ include: the convection equation (a hyperbolic PDE), where $u(x,t)$ could model fluid movement, e.g., air or some liquid, over space and time; the diffusion equation (a parabolic PDE), where $u(x,t)$ could model the temperature distribution over space and time; and the Laplace equation (an elliptic PDE), where $u(x)$ could model a steady-state diffusion equation, in the limit as $t \to \infty$.

One possible data-driven approach is to incorporate domain information by applying Eq. 1 as a "hard constraint" when training a NN on the data. This can be formulated as the following constrained optimization problem,
$$\min_\theta \mathcal{L}(u) \quad \text{s.t.} \quad \mathcal{F}(u) = 0, \tag{2}$$
where $\mathcal{L}(u)$ is the data-fit term (including initial/boundary conditions), and where $\mathcal{F}$ is a constraint on the residual of the PDE system under consideration (i.e., the "physics" knowledge in the equation itself). As mentioned before, for many practical use cases, it is not possible to derive closed form solutions for these problems, and it is often quite difficult to solve problems of the form of Eq. 2, with $\mathcal{F}(u)$ as a hard constraint.

Another (related but different) data-driven approach is to impose the constraint as a "soft constraint" on the outputs of the NN model,
$$\min_\theta \mathcal{L}(u) + \lambda_\mathcal{F} \mathcal{F}(u), \tag{3}$$
$$\mathcal{L}(u) = \mathcal{L}_{u_0} + \mathcal{L}_{u_b}. \tag{4}$$
Here, $\mathcal{L}_{u_0}$ and $\mathcal{L}_{u_b}$ measure the misfit of the NN prediction and the initial/boundary conditions (which are pre-specified/given as input to the problem), and $\theta$ denotes the NN parameters (which takes $(x,t)$, and possibly other quantities, as inputs and then outputs $u(x,t)$). Furthermore, $\lambda_\mathcal{F}$ is a regularization parameter that controls the emphasis on the PDE based residual (which we ideally want to be zero). The goal is then to use ML methodologies (stochastic optimization, etc.) to train this NN model to minimize the loss in Eq. 3. In particular, the NN is trained to minimize this modified loss function, where the modification is to penalize the violations of $\mathcal{F}(u)$ for some $\lambda_\mathcal{F} \geq 0$. However, even with a large training dataset, this approach does not guarantee that the NN will obey the conservation/governing equations in the constraint Eq. 1. In many SciML problems, these sorts of constraints on the system matter, as they correspond to physical mechanisms of the system. For example, if the conservation of energy equation is only approximately satisfied, then the system being simulated may behave qualitatively differently or even result in unrealistic solutions.

We should also note that this approach of incorporating physics-based regularization, where the regularization constraint, $\mathcal{L}_\mathcal{F}$, corresponds to a differential operator, is *very* different than incorporating much simpler norm-based regularization (such as $L_1$ or $L_2$ regularization), as is common in ML more generally. Here, the regularization operator, $\mathcal{L}_\mathcal{F}$ is non-trivially structured—it involves a differential operator that could actually be ill-conditioned, and it does not correspond to a nice convex set (as does

a norm ball). Moreover, $\mathcal{L}_{\mathcal{F}}$ corresponds to actual physical quantities, and there is often an important distinction between satisfying the constraint exactly versus satisfying the constraint approximately (the soft constraint approach doing only the latter).

**Main contributions.** The contributions of this paper are as follows:

- We analyze PINN models on simple, yet physically relevant, problems of convection, reaction, and reaction-diffusion. We find that the vanilla/regular PINN approach only works for very easy parameter regimes (i.e., small PDE coefficients), but that it fails to learn relevant physics in even moderately more challenging physical regimes, even for problems that have simple closed-form analytical solutions. For many cases, the vanilla PINN approach achieves almost 100% error, as compared to the ground truth solution, even after extensive hyperparameter tuning. (See §3 for details.)

- We analyze the loss landscape of trained PINN models and find that adding/increasing the PDE-based soft constraint regularization ($\mathcal{L}_{\mathcal{F}}$ in Eq. 3) makes it *more* complex and harder to optimize, especially for cases with non-trivial coefficients. We also study how the loss landscape changes as the regularization parameter ($\lambda_{\mathcal{F}}$) is changed. We find that reducing the regularization parameter can help alleviate the complexity of the loss landscape, but this in turn leads to poor solutions with high errors that do not satisfy the PDE/constraint. (See §4 for details.)

- We demonstrate that the NN architecture has the capacity/expressivity to find a good solution, thereby showing that these problems are not due to the limited capacity of the NN architecture. Instead, we argue that the failure is due to optimization difficulties associated with the PINN's soft PDE constraint. (See §5 for details.)

- We propose two paths forward to address these failure modes through (i) curriculum regularization and (ii) posing the learning problem as a sequence-to-sequence learning task. First, in curriculum regularization, we start by imposing the PDE constraint ($\mathcal{L}_{\mathcal{F}}$) with small coefficients, which are progressively increased to the target problem's settings as the model gets trained. This gives the NN an opportunity to first train with *easier* constraints, before it is exposed to the target constraint which could be hard to optimize from the beginning. Second, we show that changing the learning problem to a sequence-to-sequence learning problem can reduce the PINN error, again without any change to the NN architecture. In this setup, the NN is trained on a time segment, instead of the full space-time, which could be more difficult to learn. The task is then to predict the solution and reduce the loss only over smaller time segments. We extensively test both approaches and show that they can reduce the error by up to 1-2 orders of magnitude as compared to regular PINN training, and in many cases can better capture "sharp" features in the solution. (See §5 for details.)

- We have open sourced our framework [26] which is built on top of PyTorch both to help with reproducibility and also to enable other researchers to extend the results.

## 2 Related work

There is a large body of related work, and here we briefly discuss the most related lines of work.

**Machine learning and PDEs.** ML approaches for PDE problems have been increasing rapidly in recent years [13, 19]. A number of tools and methodologies now exist to solve scientific problems by combining ML and domain insights [14, 20, 27, 28, 38]. As mentioned earlier, a popular approach to combine ML and physical knowledge is to include aspects of the PDE term as part of the optimization process via regularization. A notable aspect of such an approach is that the NN can be trained only on data that comes from the governing equation(s) itself (though additional data can be included as well, if available), i.e., with a relatively small amount of data. This has garnered interest and shown successful results in a wide variety of science and engineering problems and applications [3, 11, 16, 29–31, 43].

However, there have also been issues observed with this formulation. For example, it did not work well for stiff ordinary differential equations (ODEs) describing chemical kinetics [15], for certain heterogeneous media problems [7], or for certain fluid flow problems [10]. Furthermore, PINN models have been analyzed in the context of neural tangent kernels (i.e., towards the infinite width

limit) to study their convergence [36, 37]. This work found some cases where the model failed (such as when the target function exhibits "high frequency features") and showed some preliminary solutions via the lens of the neural tangent kernel. It has been argued that some of these problems may be due an imbalance in back-propagated gradients in the loss function during training, and a learning-rate annealing scheme has been proposed to mitigate this [35].

**Physical priors and constraints in NNs.** Imposing physical priors and constraints on NN systems is common in SciML problems, as a way to try to enforce a property of interest. This idea has been introduced in different forms in the past (for instance [5, 18, 25, 28, 32]). Some approaches have focused on embedding specialized physical constraints into NNs, such as conservation of energy or momentum [4, 12] or multiscale features [34]. While methods focusing on constraining the output of the NN are more common, it is difficult to enforce such constraints exactly in ML settings. Previous work has tried to impose hard constraints in ML (both within the context of SciML and otherwise) [6, 21, 22, 24, 40], although this can be computationally expensive, and does not guarantee better results or convergence.

## 3    Possible failure modes for physics-informed neural networks

In this section, we highlight several examples where the PINN formulation defined in Eq. 3 does not predict the solution well. We first demonstrate this with two different types of simple, canonical PDE/ODE systems which have simple analytical solutions: convection ( §3.1), and reaction ( §A). We then also include a diffusion component by looking at the reaction-diffusion problem ( §3.2). Note that the convection problem has a linear PDE constraint, and reaction/reaction-diffusion problems both have non-linear PDE terms.[2] We show that PINNs can only learn simple problems with very small parameter values (e.g., small convection or reaction coefficients). We demonstrate that these models fail to learn the relevant physical phenomena for non-trivial cases (e.g., relatively larger coefficients). As we will see, while adding the physical constraint as a soft regularization may be easier to deploy and optimize with existing unconstrained optimization methods, this approach does come with trade-offs, including that in many cases the optimization problem becomes much more difficult to solve.

**Experiment setup.** We study both linear and non-linear PDEs/ODEs, and we vary the convection, reaction, and diffusion coefficients for each problem (hereafter, we refer to these as *PDE coefficients*). For each problem, we aim to minimize the loss function in Eq. 3. We use a 4-layer fully-connected NN with 50 neurons per layer, a hyperbolic tangent activation function, and randomly sample collocation points $(x, t)$ on the domain. Furthermore, all the systems that we consider have periodic boundary conditions. We enforce this through an extra term in the loss function that takes the difference between the predicted NN solution at each boundary. We train this network using the L-BFGS optimizer and sweep over learning rates from $1e-4$ to $2.0$.[3] After training the PINN, we measure the $L_2$ relative and absolute errors between the PINN's predicted solution and the analytical solution. The $L_2$ relative error is $\frac{1}{N} \sum_{i=0}^{N} \frac{\| \hat{u} - u \|_2}{\| u \|_2}$; and the absolute error is $\frac{1}{N} \sum_{i=0}^{N} \| \hat{u} - u \|_2$, where $N$ is the number of evaluation grid points, $\hat{u}$ is the predicted solution by the PINN, and $u$ is the true solution. For all cases, we run models at least ten times with different preset random seeds, and we average the relative and absolute errors in $u(x, t)$. For each loss function, $\hat{u}$ is the output of the NN and shorthand for $\hat{u} = NN(\theta, x, t)$.

---

[2]Note that for convection, this does not necessarily mean that the mapping from initial to final solution is also linear. This just means that the terms in the PDE are linear.

[3]For reasons that are only partially understood, L-BFGS methods tend to perform better for existing PINN problems. While variants of stochastic gradient descent are much more popular in computer vision, natural language processing, and recommendation systems, we found that they underperform in comparison to L-BFGS.

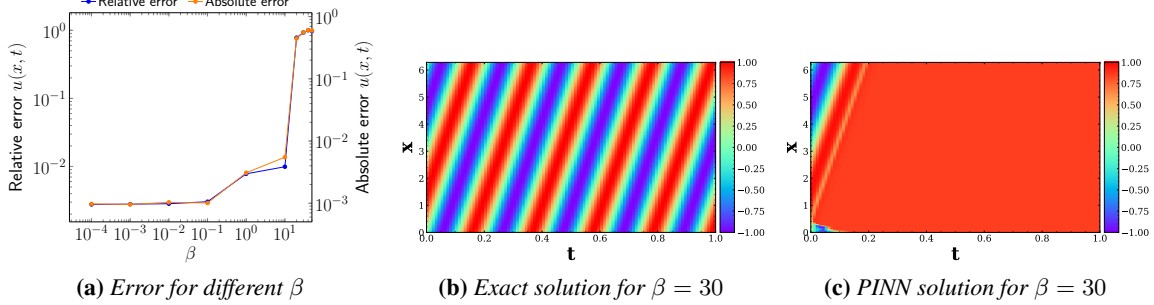

**(a)** *Error for different $\beta$*    **(b)** *Exact solution for $\beta = 30$*    **(c)** *PINN solution for $\beta = 30$*

**Figure 1:** *Prediction error for 1D convection ( §3.1) problem, when $\beta$ is changed. The PINN has difficulty predicting the solution past a certain timestep, but is able to fit the boundary conditions. Additional figures for different $\beta$ values can be seen in Fig. C.1.*

## 3.1  Learning convection

**Problem formulation.**  We first consider a one-dimensional convection problem, a hyperbolic PDE which is commonly used to model transport phenomena:

$$\frac{\partial u}{\partial t} + \beta \frac{\partial u}{\partial x} = 0, \quad x \in \Omega, \ t \in [0, T], \tag{5}$$

$$u(x, 0) = h(x), \quad x \in \Omega.$$

Here, $\beta$ is the convection coefficient and $h(x)$ is the initial condition. For constant $\beta$ and periodic boundary conditions, this problem has a simple analytical solution:

$$u_{\text{analytical}}(x, t) = F^{-1}\big(F(h(x))e^{-i\beta kt}\big), \tag{6}$$

where $F$ is the Fourier transform, $i = \sqrt{-1}$, and $k$ denotes frequency in the Fourier domain. The general loss function for this problem (corresponding to Eq. 3) is

$$\mathcal{L}(\theta) = \frac{1}{N_u} \sum_{i=1}^{N_u} \Big(\hat{u} - u_0^i\Big)^2 + \frac{1}{N_f} \sum_{i=1}^{N_f} \lambda_i \Big(\frac{\partial \hat{u}}{\partial t} + \beta \frac{\partial \hat{u}}{\partial x}\Big)^2 + \mathcal{L}_{\mathcal{B}}, \tag{7}$$

where $\hat{u} = NN(\theta, x, t)$ is the output of the NN, and $\mathcal{L}_{\mathcal{B}}$ is the boundary loss. For periodic boundary conditions with $\Omega = [0, 2\pi)$, this loss is:

$$\mathcal{L}_{\mathcal{B}} = \frac{1}{N_b} \sum_{i=1}^{N_b} \Big(\hat{u}(\theta, 0, t) - \hat{u}(\theta, 2\pi, t)\Big)^2. \tag{8}$$

We use the following simple initial and periodic boundary conditions:

$$u(x, 0) = sin(x), \tag{9}$$
$$u(0, t) = u(2\pi, t).$$

**Observations.**  We apply the PINN's soft regularization to this problem, and we optimize the loss function in Eq. 7. After training, we measure the relative and absolute errors between the PINN's predicted solution and the analytical solution, as reported in Fig. 1(a). As one can see, the PINN is only able to achieve good solutions for small values of convection coefficient, and it fails when $\beta$ becomes larger, reaching a relative error of almost 100% for $\beta > 10$. We also provide visualization of the exact and PINN solution in Fig. 1(b-c). One can clearly see that the PINN is unable to learn the solution. As we will later show, the NN architecture does have enough capacity to find the solution, but the training/optimization problem is very difficult to solve with PINNs (and importantly, it may require extensive hyperparameter tuning which is often not feasible in practice).

## 3.2  Learning reaction-diffusion

**Problem formulation.**  We next look at a reaction-diffusion system, where we add a diffusion operator to the reaction equation discussed above. Note that for pure diffusion, the solution dissipates

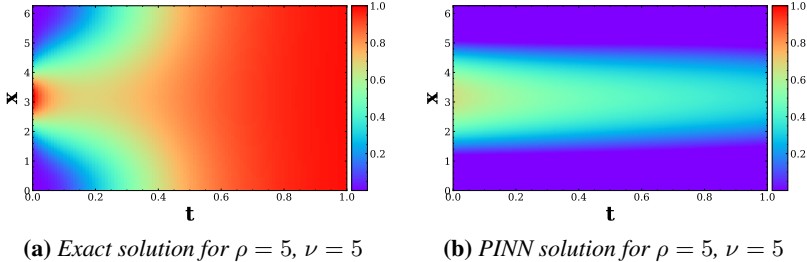

**(a)** *Exact solution for $\rho = 5$, $\nu = 5$*  **(b)** *PINN solution for $\rho = 5$, $\nu = 5$*

**Figure 2:** *Prediction error for 1D reaction-diffusion ( §3.2) problem. We can clearly see that the PINN has difficulty predicting the solution (especially the "sharpness" of the solution) and is unable to capture the correct behavior. Additional figures for different $\nu$ values can be seen in Fig. D.1.*

to a steady-state of uniform/constant distribution, which may be trivial to learn. Therefore, we consider studying the reaction-diffusion system:

$$\frac{\partial u}{\partial t} - \nu \frac{\partial^2 u}{\partial x^2} - \rho u(1-u) = 0, \quad x \in \Omega, \ t \in (0, T], \tag{10}$$

$$u(x, 0) = h(x), \quad x \in \Omega.$$

Here, $\nu$ ($\nu > 0$) is the diffusion coefficient. The solution of such a system can be solved for via Strang splitting, i.e., splitting the equation into two separate models (a reaction component and a diffusion component):

$$\frac{du}{dt} = \rho u(1-u)$$

$$\frac{du}{dt} = \nu \frac{\partial^2 u}{\partial x^2}. \tag{11}$$

For each timestep, we can solve the reaction equation through Eq. 15 (in §A). The diffusion equation has the following analytical solution:

$$u_{\text{analytical}}(x, t) = F^{-1}\big(F(u(x, t = t^n))e^{-\nu k^2 t}\big), \tag{12}$$

where $u(x, t = t^n)$ is the solution at the $n^{th}$ time step. We solve the reaction equation for each timestep, and then use the reaction solution as the initial condition to solve the diffusion component and get the final solution.

The general loss function for this problem is,

$$\mathcal{L}(\theta) = \frac{1}{N_u} \sum_{i=1}^{N_u} \left(\hat{u} - u_0^i\right)^2 +$$

$$\frac{1}{N_f} \sum_{i=1}^{N_f} \lambda_i \left(\frac{\partial \hat{u}}{\partial t} - \nu \frac{\partial^2 \hat{u}}{\partial x^2} - \rho \hat{u}(1-\hat{u})\right)^2 + \mathcal{L}_{\mathcal{B}}, \tag{13}$$

where $\mathcal{L}_{\mathcal{B}}$ is the boundary loss. Similar to the previous example, periodic boundary conditions can be enforced by including $\mathcal{L}_{\mathcal{B}}$ from Eq. 8 as an extra term in the loss.

**Observations.** Similar to the previous case, we can see that the PINN also fails to learn reaction-diffusion. We illustrate a case in Fig. 2 with $\rho = 5$, when $\nu = 5$. The PINN achieves a high relative error of 93%. Here, we can clearly see that the PINN is unable to capture either the reaction or diffusion component. Additional figures for different $\nu$ values can be seen in Fig. D.1. In particular, for $\nu = 2$ the PINN achieves a relative error of 50%. Here, we see that it is unable to capture the "sharper" transitions, though it can predict the center of the solution a little better.

## 4 Diagnosing possible failure modes for physics-informed NNs

Thus far, we have shown that PINNs can result in high errors even for simple physical regimes, in particular for PDEs/ODEs with non-trivial convection/reaction/diffusion coefficients. Here, we

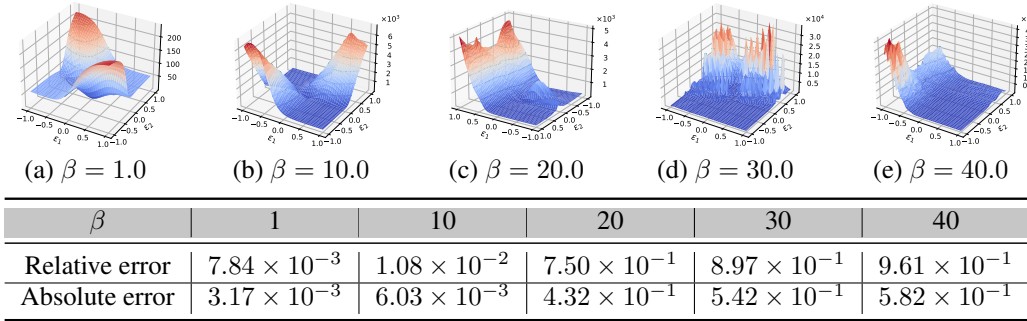

| $\beta$ | 1 | 10 | 20 | 30 | 40 |
|---|---|---|---|---|---|
| Relative error | $7.84 \times 10^{-3}$ | $1.08 \times 10^{-2}$ | $7.50 \times 10^{-1}$ | $8.97 \times 10^{-1}$ | $9.61 \times 10^{-1}$ |
| Absolute error | $3.17 \times 10^{-3}$ | $6.03 \times 10^{-3}$ | $4.32 \times 10^{-1}$ | $5.42 \times 10^{-1}$ | $5.82 \times 10^{-1}$ |

**Figure 3:** *Loss landscapes for varying values of $\beta$, for the 1D convection example in §3.1. The loss landscape is more smooth at low $\beta$, and it becomes increasingly more complex as $\beta$ increases, which can make the optimization problem more difficult. In particular, at higher $\beta$, the optimizer gets stuck in a certain regime. These results support that adding the PDE soft regularization term results in a more complex optimization loss landscape.*

demonstrate that one of the underlying reasons for this arises due to the PDE-based soft constraint of $\mathcal{L}_{\mathcal{F}}$, which makes the loss landscape difficult to optimize. We first (in §4.1) analyze the loss landscape to illustrate how increasing this soft regularization can lead to more complex loss landscapes, thus leading to optimization difficulties. We then (in §B) demonstrate how this is related to regularizing with differential operators, which can result in ill-conditioning.

### 4.1 Soft PDE regularization and optimization difficulties

Here, we analyze how the loss landscape changes for different regimes for the convection problem in §3.1 with/without the soft regularization in PINNs. We show that adding the soft regularization can actually make the problem harder to optimize, i.e., the regularization leads to less smooth loss landscapes. For all the experiments, we plot the loss landscape by perturbing the (trained) model across the first two dominant Hessian eigenvectors and computing the corresponding loss values. This tends to be more informative than perturbing the model parameters in random directions [41, 42].

Figure 3 shows the loss landscape for the convection problem (discussed in §3.1), for different $\beta$ values. Interestingly, the loss landscape at a relatively low $\beta = 1$ is rather smooth, but increasing $\beta$ further results in a complex and non-symmetric loss landscape. It is also evident that the optimizer has gotten stuck in a local minima with a very high loss function for large $\beta$ values.

Finally, we study the impact of changing the weight/multiplier for the soft regularization term (i.e., the $\lambda$ parameter in Eq. 3), which can be relevant in improving PINN performance [35]. While we find that tuning $\lambda$ can help change the error, it cannot resolve the problem, as shown in Fig. E.1. Note that as the regularization parameter is increased, the loss landscape becomes increasingly more complex and harder to optimize (additionally, see the z-axis scale).

## 5 Expressivity versus optimization difficulty

In this section, we first show that the failure modes we observed are not necessarily due to the specific NN architecture that we used in our experiments. In particular, we show that the NN model does have the expressivity/capacity to learn the convection/reaction/diffusion coefficient cases where the vanilla PINN method fails. Additionally, in the process of demonstrating this, we also describe two methods that lead to significantly lower error rates. In particular, we show that changing the learning paradigm to *curriculum regularization* can make the optimization problem easier to solve (as discussed in §5.1). Second, we show that posing the problem as *sequence-to-sequence learning* may lead to better results than learning the entire state-space at once (as discussed in §5.2).

### 5.1 Curriculum PINN Regularization

One may contend that the failure modes shown in §3 may be because the NN does not have enough capacity. Here, we show that this is not the underlying reason. To do so, we devise a "curriculum

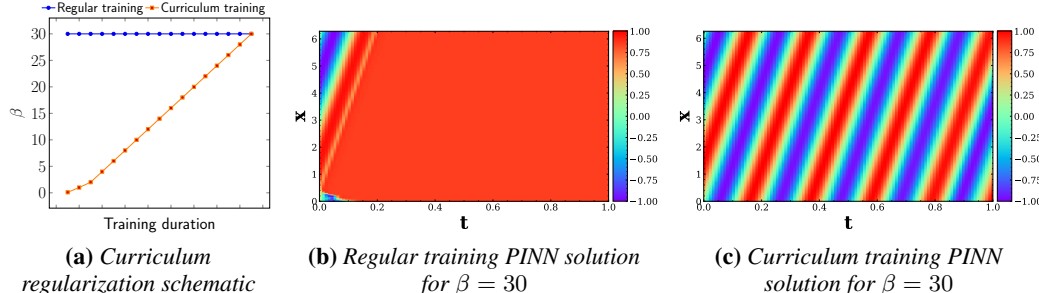

**(a)** *Curriculum regularization schematic*

**(b)** *Regular training PINN solution for $\beta = 30$*

**(c)** *Curriculum training PINN solution for $\beta = 30$*

**Figure 4:** *Schematic outlining curriculum regularization and example result for 1D convection from §3.1 The training procedure for regular PINNs training versus curriculum PINN training for the convection example in §3.1. The regular PINN training only involves training at $\beta = 30$, while curriculum regularization starts at a lower $\beta$, trains a model, and then uses the weights of this model to reinitialize the NN for training the next $\beta$. The curriculum training approach is able to do significantly better (by almost two orders of magnitude).*

| | | Regular PINN | Curriculum training |
|---|---|---|---|
| 1D convection: $\beta = 20$ | Relative error | $7.50 \times 10^{-1}$ | $\mathbf{9.84 \times 10^{-3}}$ |
| | Absolute error | $4.32 \times 10^{-1}$ | $\mathbf{5.42 \times 10^{-3}}$ |
| 1D convection: $\beta = 30$ | Relative error | $8.97 \times 10^{-1}$ | $\mathbf{2.02 \times 10^{-2}}$ |
| | Absolute error | $5.42 \times 10^{-1}$ | $\mathbf{1.10 \times 10^{-2}}$ |
| 1D convection: $\beta = 40$ | Relative error | $9.61 \times 10^{-1}$ | $\mathbf{5.33 \times 10^{-2}}$ |
| | Absolute error | $5.82 \times 10^{-1}$ | $\mathbf{2.69 \times 10^{-2}}$ |

**Table 1:** *Training the PINN gradually on more difficult problems improves performance. 1D convection example in §3.1. The curriculum training approach achieves significantly better errors.*

regularization" method to warm start the NN training by finding a good initialization for the weights. Instead of training the PINN to learn the solution right away for cases with higher $\beta/\rho$, we start by training the PINN on lower $\beta/\rho$ (easier for the PINN to learn) and then gradually move to training the PINN on higher $\beta/\rho$, respectively. We test these results for the examples in §3.1 and §A. This is somewhat analogous to curriculum learning in ML [1], but applied by progressively making the PDE/ODE harder to solve.

Figure 4 shows the training procedure for an example convection case ( §3.1) with $\beta = 30$. As Fig. 4(c) shows, the curriculum regularization approach results in a much more accurate solution than regular PINN training. With curriculum regularization, the relative error is almost two orders of magnitude lower. Additionally, this is true across all the other regimes that we found regular PINNs to fail, as shown in Tab. 1. In Fig. E.2, we also show that curriculum regularization not only decreases error significantly, but also decreases the variance of the error. In Fig. E.3, we see that curriculum regularization results in a much smoother loss landscape as compared to regular PINN training.

Curriculum regularization also works well for the reaction example in §A. In this case, we start by training with a low $\rho$ value (reaction coefficient), and then increase gradually to higher $\rho$ values. The results can be seen in Fig. E.4. We can see that the error is 0.1 - 0.6 orders of magnitude lower for $\rho = 2 - 4$ (when the regular PINN error is not as high), and then greatly decreases error by 1-2 orders of magnitude for $\rho = 5 - 10$. As we discussed before, PINN has difficulty in learning sharp features for high values of $\rho$. However, the curriculum regularization overcomes this, even for $\rho = 10$, as seen in Fig. E.4(c).

### 5.2 Sequence-to-sequence learning vs learning the entire space-time solution

The original PINN approach of [28] trains the NN model to predict the entire space-time at once (i.e., predict $u$ for all locations and time points). In certain cases, this can be more difficult to learn. Here, we demonstrate that it may be better to pose the problem as a sequence-to-sequence (seq2seq) learning task, where the NN learns to predict the solution at the next time step, instead of all times.

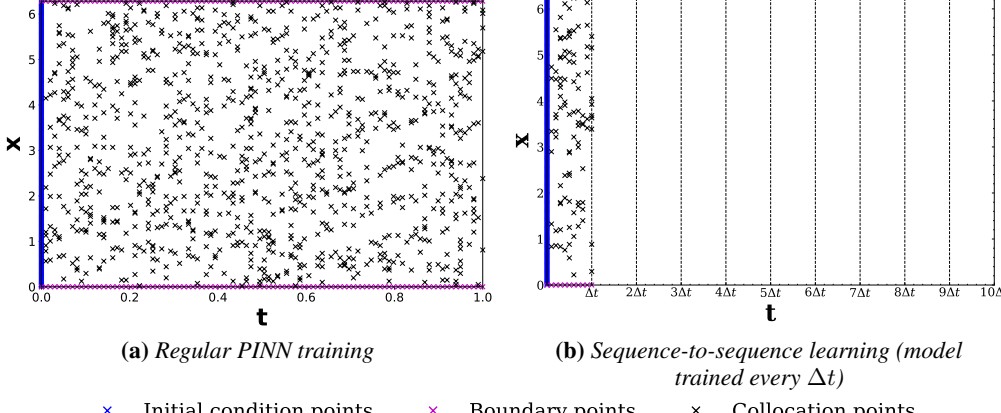

**(a)** *Regular PINN training*          **(b)** *Sequence-to-sequence learning (model trained every $\Delta t$)*

×   Initial condition points          ×   Boundary points          ×   Collocation points

**Figure 5:** *Schematic outlining seq2seq learning. In contrast to regular PINN training, the solution in seq2seq learning is predicted for only one $\Delta t$ step at a time. Then, the predicted solution at $t = \Delta t$ is used as the initial condition for the next segment. To allow fair comparison, we keep the total number of collocation points to be exactly the same in either approach. That is, we do not increase the number of collocation points for seq2seq learning in the right, and keep it to be the same as in the corresponding segment in the left figure.*

|  |  | Entire state space | $\Delta t = 0.05$ | $\Delta t = 0.1$ |
|---|---|---|---|---|
| $\nu = 2, \rho = 5$ | Relative error | $5.07 \times 10^{-1}$ | $2.04 \times 10^{-2}$ | $\mathbf{1.18 \times 10^{-2}}$ |
|  | Absolute error | $2.70 \times 10^{-1}$ | $1.06 \times 10^{-2}$ | $\mathbf{6.41 \times 10^{-3}}$ |
| $\nu = 3, \rho = 5$ | Relative error | $7.98 \times 10^{-1}$ | $1.92 \times 10^{-2}$ | $\mathbf{1.56 \times 10^{-2}}$ |
|  | Absolute error | $4.79 \times 10^{-1}$ | $1.01 \times 10^{-2}$ | $\mathbf{8.17 \times 10^{-3}}$ |
| $\nu = 4, \rho = 5$ | Relative error | $8.84 \times 10^{-1}$ | $2.37 \times 10^{-2}$ | $\mathbf{1.59 \times 10^{-2}}$ |
|  | Absolute error | $5.74 \times 10^{-1}$ | $1.15 \times 10^{-2}$ | $\mathbf{8.01 \times 10^{-3}}$ |
| $\nu = 5, \rho = 5$ | Relative error | $9.35 \times 10^{-1}$ | $\mathbf{2.36 \times 10^{-2}}$ | $2.39 \times 10^{-2}$ |
|  | Absolute error | $6.46 \times 10^{-1}$ | $\mathbf{1.09 \times 10^{-2}}$ | $1.15 \times 10^{-2}$ |
| $\nu = 6, \rho = 5$ | Relative error | $9.60 \times 10^{-1}$ | $2.81 \times 10^{-2}$ | $\mathbf{2.69 \times 10^{-2}}$ |
|  | Absolute error | $6.84 \times 10^{-1}$ | $\mathbf{1.17 \times 10^{-2}}$ | $1.28 \times 10^{-2}$ |

**Table 2:** *Predicting the entire state space versus discretizing the state space (i.e., seq2seq learning) for 1D reaction-diffusion ( §3.2). The seq2seq learning achieves lower error for both $\Delta t = 0.05$ and $\Delta t = 0.1$, in comparison to the PINN's approach of predicting the entire state space at once.*

This way, we can use a marching-in-time scheme to predict different sequences/time points. Note that the only data available here is from the PDE itself, i.e., just the initial condition. We take the prediction at $t = \Delta t$ and use this as the initial condition to make a prediction at $t = 2\Delta t$, and so on. This is schematically outlined in Fig. 5.

We test this scheme by using the exact same NN architecture as in previous sections, and we report the results in Tab. E.1 for the convection problem of §3.1, Tab. E.2 for the reaction problem of §A, and Tab. 2 for the reaction-diffusion problem of §3.2. We compare the relative/absolute error when the learning is posed as a seq2seq problem (i.e., predicting the state space with a "time marching scheme" of one timestep prediction at a time) to the PINN approach of predicting the whole state space at once.[4]

We explore the following cases where the PINN does poorly, varying $\beta$, $\rho$, and $\nu$ coefficients:

---

[4]To have a fair comparison between the two methods (time marching versus predicting entire state space at once), for the time marching method, we use the same number of collocation (interior) points for both. For example, for T = [0, 1], if we use 1000 collocation points to predict the entire state space, then for $\Delta t = 0.1$ we use 100 collocation points per section.

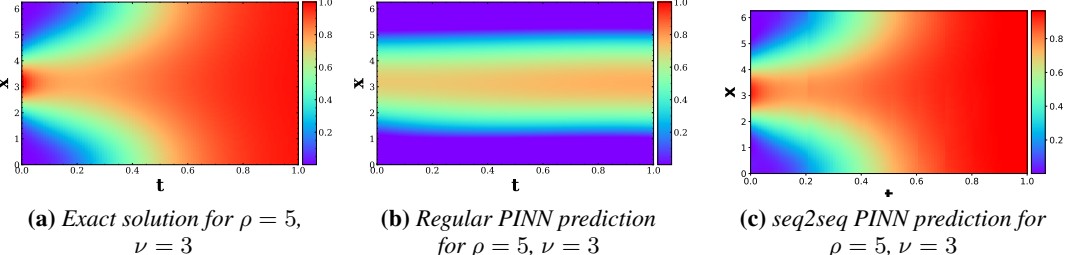

**(a)** *Exact solution for $\rho = 5$, $\nu = 3$*

**(b)** *Regular PINN prediction for $\rho = 5$, $\nu = 3$*

**(c)** *seq2seq PINN prediction for $\rho = 5$, $\nu = 3$*

**Figure 6:** *Predicting the entire state space vs seq2seq learning for 1D reaction-diffusion. The regular PINN is unable to capture the "sharp" and/or diffusive features correctly. However, the seq2seq learning approach is able to capture the correct solution, and achieves almost two orders of magnitude lower error.*

1) For 1D convection ( §3.1), higher $\beta$ values from 30-40.

2) For 1D reaction ( §A), $\rho$ coefficients from 5-10.

3) For 1D reaction-diffusion ( §3.2), a fixed $\rho = 5$ and $\nu$ coefficients from 2-6.

For these cases, we find that posing the problem as seq2seq learning results in significantly lower error. The difference is particularly striking for the reaction and reaction-diffusion cases, where the seq2seq PINN model decreases error by almost two orders of magnitude. An example case is shown Fig. 6, where the seq2seq approach is able to recover the solution, while regular PINNs does very poorly. Note that this behavior also has analogues with numerical methods used in scientific computing, where space-time problems are typically harder to solve, as compared to time marching methods [8]. Intuitively, since the problem is ill-conditioned, restricting the dimensions is expected to help. Furthermore, the underlying function/mapping of the input to the solution should be much simpler to approximate over a smaller time span, as compared to the full time horizon.

These initial results are promising, and further developments may lead to still better ways of using PINNs and learning PDEs. In particular, using more sophisticated methods to predict timesteps across the state space may provide improved performance, as may including more sophisticated seq2seq approaches and tuning the regularization parameter (i.e., amount of constraint added).

## 6    Conclusions

PINNs—and SciML more generally—hold great promise for expanding the scope of ML methodology to important problems in science and engineering. For these problems, however, integrating ML methods with PDE-based domain-driven constraints as a soft regularization term can lead to subtle and critical issues. In particular, we show that this approach can have fundamental limitations which results in failure modes for learning relevant physics commonly used in different fields of science. To show this, we picked two fundamental PDE problems of diffusion and convection and showed that the PINN only works for very simple cases, failing to learn the relevant physical phenomena for even moderately more challenging regimes. We then analyzed the problem to characterize the underlying reasons why these failures occur. In particular, we studied the PINN loss landscape behavior and found it becomes it becomes increasingly complex for large values of diffusion or convection coefficients, and with/without non-homogeneous forcing. We also discussed that the problem is not necessarily due to the limited capacity of the NN, but that it is partly an optimization problem resulting in the PDE-based soft constraint used in PINNs. Furthermore, we showed that the PINN approach of solving for the entire space-time at once may not be efficient, and instead posing the problem as a sequence-to-sequence learning task can provide lower error rates. Addressing these and related issues will be critical if we hope to go beyond existing cut-and-paste approaches, toward engineering a more intimate connection between scientific methodologies and ML methodologies. This will be needed to deliver on the promise of PINNs and SciML more generally.

# 7  Acknowledgements.

We are thankful to Shashank Subramanian for his feedback and contributions. We also acknowledge helpful discussions with Prof. George Biros, Geoffrey Negiar, and Daniel Rothchild. ASK was supported by Laboratory Directed Research and Development (LDRD) funding under Contract Number DE-AC02-05CH11231 at LBNL and the Alvarez Fellowship in the Computational Research Division at LBNL. AG was supported through funding from Samsung SAIT. MWM would also like to acknowledge the UC Berkeley CLTC, ARO, NSF, and ONR. The UC Berkeley team also acknowledges gracious support from Intel corporation, Intel VLAB, Samsung, Amazon AWS, Google Cloud, Google TPU Research Cloud, and Google Brain (in particular Prof. David Patterson, Dr. Ed Chi, and Jing Li). Our conclusions do not necessarily reflect the position or the policy of our sponsors, and no official endorsement should be inferred.

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
