

**Figure A.1:** *Prediction error for 1D reaction ( §A) problem, when $\rho$ is changed. The PINN has difficulty predicting any part of the solution (and can't even predict the first few timesteps), and only predicts a homogeneous solution everywhere.*

# A    Learning reaction

We include an additional example studying the reaction case, to complement the additional studies in §3.

**Problem formulation.**    We consider an example of a one-dimensional reaction equation, which is commonly used to model chemical reactions. We look at the reaction term in Fisher's equation, which is a semi-linear ordinary differential equation:

$$\frac{\partial u}{\partial t} - \rho u(1 - u) = 0, \quad x \in \Omega, \ t \in (0, T],$$

$$u(x, 0) = h(x), \quad x \in \Omega.$$

(14)

Here, $\rho$ is the reaction coefficient and $h(x)$ is the initial condition. The reaction problem has a simple analytical solution for periodic boundary conditions and constant $\rho$:

$$u_{\text{analytical}}(x, t) = \frac{h(x)e^{\rho t}}{h(x)e^{\rho t} + 1 - h(x)}.$$

(15)

The general loss function for this problem is

$$\mathcal{L}(\theta) = \frac{1}{N_u} \sum_{i=1}^{N_u} \left( \hat{u} - u_0^i \right)^2 + \frac{1}{N_f} \sum_{i=1}^{N_f} \lambda_i \left( \frac{\partial \hat{u}}{\partial t} - \rho \hat{u}(1 - \hat{u}) \right)^2 + \mathcal{L}_{\mathcal{B}},$$

(16)

where $\mathcal{L}_{\mathcal{B}}$ is the boundary loss (same as in Eq. 8). We consider the following initial and boundary conditions:

$$u(x, 0) = e^{-\frac{(x-\pi)^2}{2(\pi/4)^2}},$$

$$u(0, t) = u(2\pi, t).$$

(17)

**Observations.**    We report the relative/absolute error of the PINN with respect to the ground truth in Fig. A.1. Similar to the convection case, we can see that the PINN is only able to learn the problem for very small values of the reaction coefficient, $\rho$. However, the error quickly gets to 100% as $\rho$ is increased. The example heatmap in Fig. A.1(c) shows that the PINN is not able to predict the solution at all, and instead, it predicts a mostly homogeneous solution, close to zero, everywhere.

# B    A PDE perspective on ill-conditioned regularization

One of the difficulties with PINNs arises from the soft regularization term that includes differential operators. This term is quite different from norm-based regularization that is more common in ML, and this can actually make the problem more ill-conditioned (or less regularized). From a PDE

perspective, this is not surprising as the PDE-based regularization operator (i.e., $\mathcal{L}_{\mathcal{F}}$ term) in the PINN can in fact be ill-conditioned, which can lead to unstable numerical behavior. For example, in numerical PDE analysis, it is well-known that the condition number for the regularization operator for the diffusion problem is $\mathcal{O}(\nu N^2)^2$, where $N$ is the grid size (see §B.1 for the derivation). Similarly, the condition number for the convection problem scales as $\mathcal{O}(\beta N)^2$, which is still quite high. As such, it is not surprising that this ill-conditioned property would lead to instability which can manifest itself in large gradients and/or poor convergence behavior. (Similar results were also reported in [35].) We should however emphasize that the condition number is only one of the many factors involved in this problem, and other factors such as the complexity of the function that we are trying to approximate, the non-convex loss landscape, and the limitations of optimization algorithms need to be considered. To give an example, a highly diffusive PDE with a very large diffusion coefficient quickly results in a solution that approaches uniform/constant distribution, which can be very easy to approximate with trivial initial conditions. However, as with other ill-conditioned operators, it is in the presence of noise, or cases with non-trivial physics, that the instability appears.

### B.1 Approximate condition number scaling for PDE regularization in PINNs

We provide a derivation to obtain an approximate condition number for the PINN regularization term. First, note that the condition number of an operator is a metric for the amount that a function's output changes for a change in its input. Using this definition, we can obtain an approximate bound on the condition number for the PINN regularization operator. We should emphasize that the non-linear nature of the regularization makes the exact condition number behavior dependent on the state variable and its derivatives, and so the results obtained below are approximate.

**Convection problem.** For the convection problem, the PDE-based regularization operator is:

$$f = \frac{du}{dt} + \beta \frac{\partial u}{\partial x}. \tag{18}$$

For a small change in $\delta u$ to input, the output will change as follows:

$$\delta f = \frac{d\delta u}{dt} + \beta \frac{\partial \delta u}{\partial x}. \tag{19}$$

We can estimate that the maximum change in output (denoted as $\delta f$) is proportional to the sum of the maximum change in the first time derivative, which scales as $\mathcal{O}(\delta t^{-1})$, and the second term, which scales as $\mathcal{O}(\beta h^{-1})$. Here, $\delta t$ is the time step size and $h = 1/N$ is the grid size spacing for a uniformly discretized grid of size $N$. Assuming that the time step size is not very small, the condition number will be proportional to the second term and will scale as $\mathcal{O}(\beta N)$. Since the PINN regularization term uses $L_2$ loss, this change will be quadratically scaled.

**Diffusion problem.** Similarly, we can obtain an approximate scaling of the condition number for the diffusion case where the regularization function has the following form:

$$f = \frac{du}{dt} - \nu \frac{\partial^2 u}{\partial x^2}. \tag{20}$$

The corresponding change in the output, for a perturbation in $u$, is then:

$$\delta f = \frac{d\delta u}{dt} - \nu \frac{\partial^2 \delta u}{\partial x^2}. \tag{21}$$

Similar to the previous case, we can estimate that the maximum change in output (denoted as $\delta f$) is proportional to the sum of the maximum change in the first time, which again scales as $\mathcal{O}(\delta t^{-1})$, and the second term, which scales as $\mathcal{O}(\nu h^{-2})$. Again, assuming that the time step size is not very small, the condition number will be proportional to the second term and will scale as $\mathcal{O}(\nu N^2)$. Similarly, since the PINN regularization term uses $L_2$ loss, this change will be quadratically scaled.

Also, we observed that error increases more rapidly when $\nu$ increases (for the diffusion case), in contrast to the slower error increase with respect to $\beta$ for the convection case. Finally, as our estimates

of the condition number show, the condition number for diffusion scales as $N^2$ while convection scales as $N$.

# C   Learning convection

We include additional heatmaps for learning convection ( §3.1).

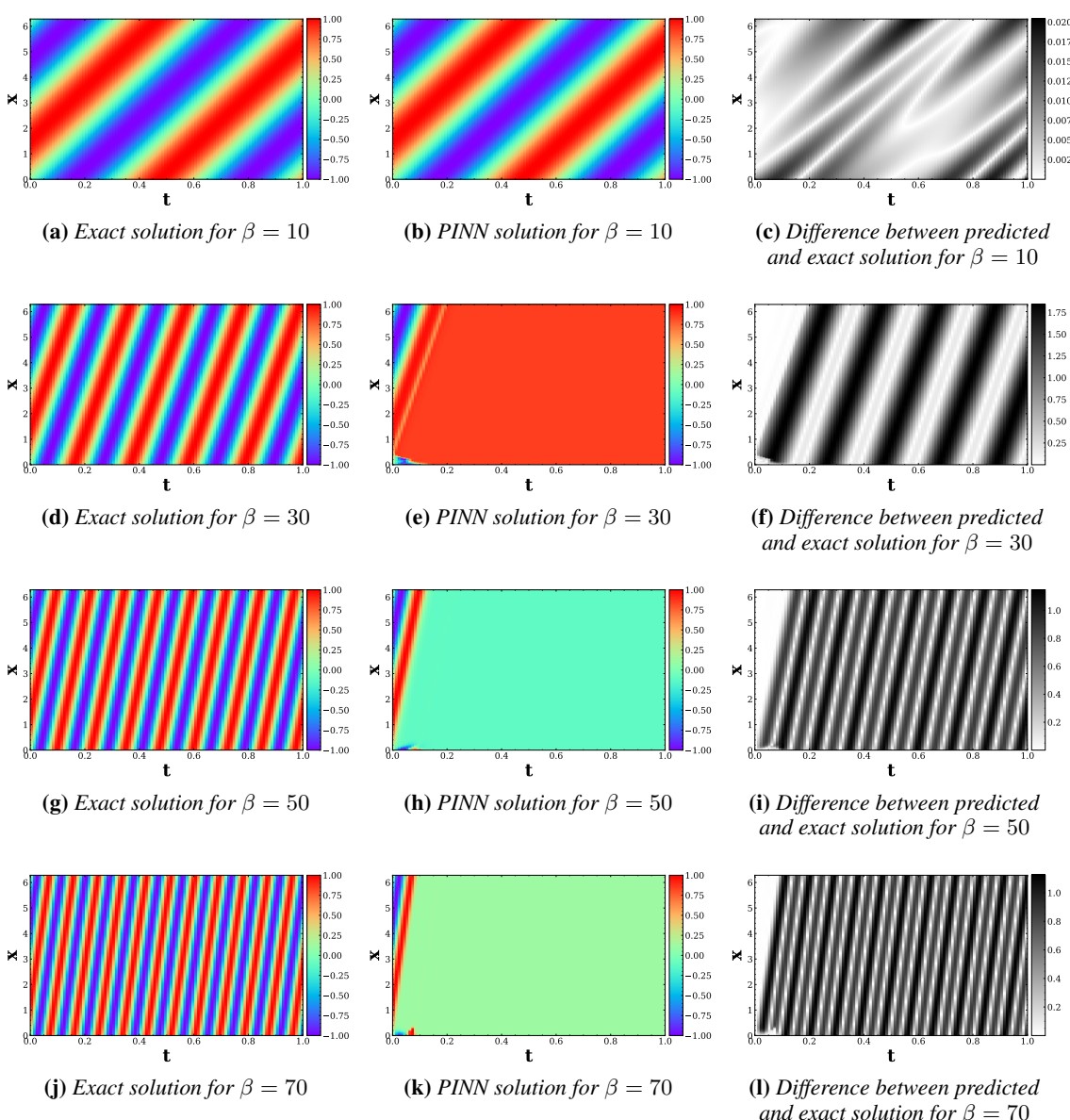

(a) *Exact solution for $\beta = 10$*

(b) *PINN solution for $\beta = 10$*

(c) *Difference between predicted and exact solution for $\beta = 10$*

(d) *Exact solution for $\beta = 30$*

(e) *PINN solution for $\beta = 30$*

(f) *Difference between predicted and exact solution for $\beta = 30$*

(g) *Exact solution for $\beta = 50$*

(h) *PINN solution for $\beta = 50$*

(i) *Difference between predicted and exact solution for $\beta = 50$*

(j) *Exact solution for $\beta = 70$*

(k) *PINN solution for $\beta = 70$*

(l) *Difference between predicted and exact solution for $\beta = 70$*

**Figure C.1:** ***Heatmap of exact vs predicted solution for 1D convection ( §3.1).*** *Heatmap of the exact solutions to the 1D convection equation, Eq. 5, for a variety of $\beta$ values and the respective PINN predicted solution. The PINN is unable to predict the solution as $\beta$ increases, past a certain timestep.*

# D Learning reaction-diffusion

We include additional heatmaps for learning reaction-diffusion ( §3.2).

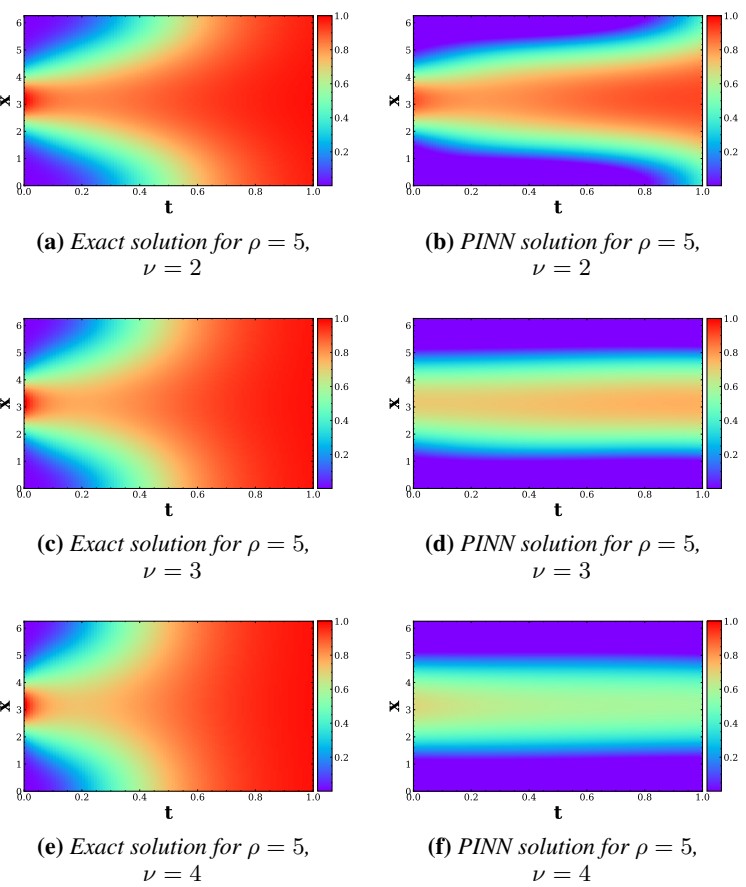

**(a)** *Exact solution for $\rho = 5$, $\nu = 2$*

**(b)** *PINN solution for $\rho = 5$, $\nu = 2$*

**(c)** *Exact solution for $\rho = 5$, $\nu = 3$*

**(d)** *PINN solution for $\rho = 5$, $\nu = 3$*

**(e)** *Exact solution for $\rho = 5$, $\nu = 4$*

**(f)** *PINN solution for $\rho = 5$, $\nu = 4$*

**Figure D.1:** *Heatmap of exact vs predicted solution for 1D reaction-diffusion ( §3.2). Heatmap of the exact solutions to the 1D reaction-diffusion equation, Eq. 10, for different $\nu$ values and the respective physics-informed NN predicted solution. The PINN is unable to predict the solution, including both capturing the "sharp" features and/or diffuse features.*

# E    Extra Results

## E.1    Extra results for loss landscapes when varying the $\lambda$ parameter

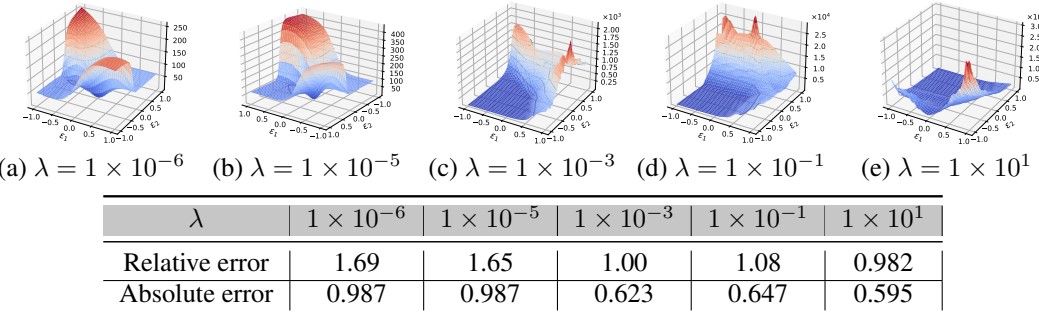

(a) $\lambda = 1 \times 10^{-6}$    (b) $\lambda = 1 \times 10^{-5}$    (c) $\lambda = 1 \times 10^{-3}$    (d) $\lambda = 1 \times 10^{-1}$    (e) $\lambda = 1 \times 10^{1}$

| $\lambda$ | $1 \times 10^{-6}$ | $1 \times 10^{-5}$ | $1 \times 10^{-3}$ | $1 \times 10^{-1}$ | $1 \times 10^{1}$ |
|---|---|---|---|---|---|
| Relative error | 1.69 | 1.65 | 1.00 | 1.08 | 0.982 |
| Absolute error | 0.987 | 0.987 | 0.623 | 0.647 | 0.595 |

**Figure E.1:** *Loss landscapes when varying the $\lambda$ parameter in $\mathcal{F}$, for the 1D convection equation in §3.1. In this example, $\beta = 30$, which is a point at which the error is high. The loss landscape becomes more complex as $\lambda$ is increased, i.e., as the regularization term grows. However, error stays consistently high (although it decreases a little as $\lambda$ is increased).*

## E.2    Extra curriculum regularization results

We include extra curriculum regularization results for §3.1 in Fig. E.2. These results demonstrate that curriculum regularization not only decreases the error in the solution, but also decreases the variance in the error.

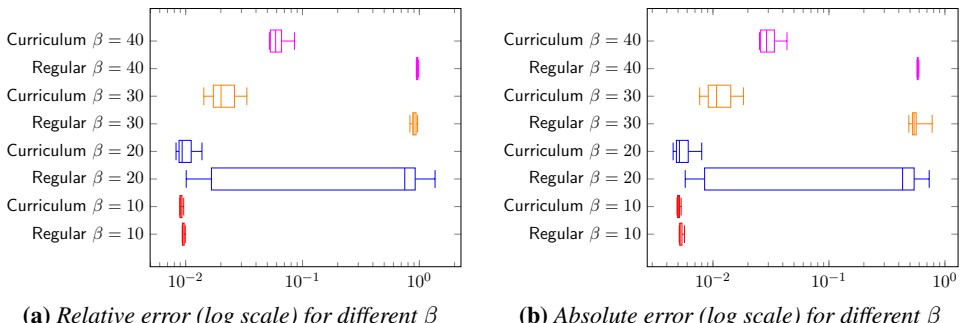

**(a)** *Relative error (log scale) for different $\beta$*    **(b)** *Absolute error (log scale) for different $\beta$*

**Figure E.2:** *Training the PINN gradually on more difficult problems improves performance. 1D convection example in §3.1. Summary of performance across 10 preset random seeds (with lowest error per seed) to show the variance in error. The curriculum learning approach achieves significantly better errors, as well as lower variance in the error.*

We include extra curriculum regularization results for the convection example in §3.1 in Fig. E.3, showing how the loss landscape becomes smoother with the curriculum regularization approach. We also include extra curriculum regularization results for the reaction example in §A in Fig. E.4. These results further demonstrate that curriculum regularization greatly decreases error in the solution.

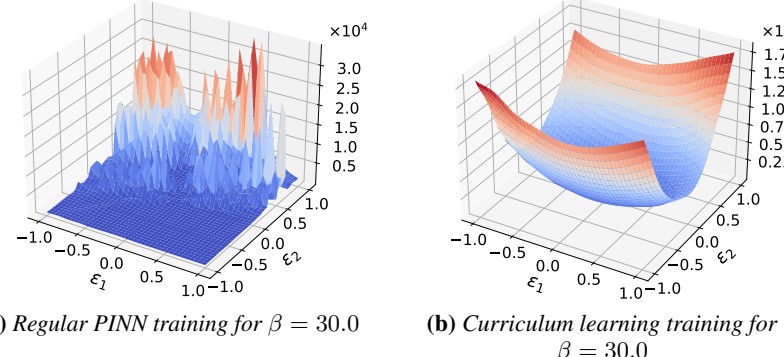

**(a)** *Regular PINN training for $\beta = 30.0$*  **(b)** *Curriculum learning training for $\beta = 30.0$*

**Figure E.3:** *Loss landscapes for regular PINN training versus curriculum learning training. The loss landscape is much smoother for the curriculum learning approach.*

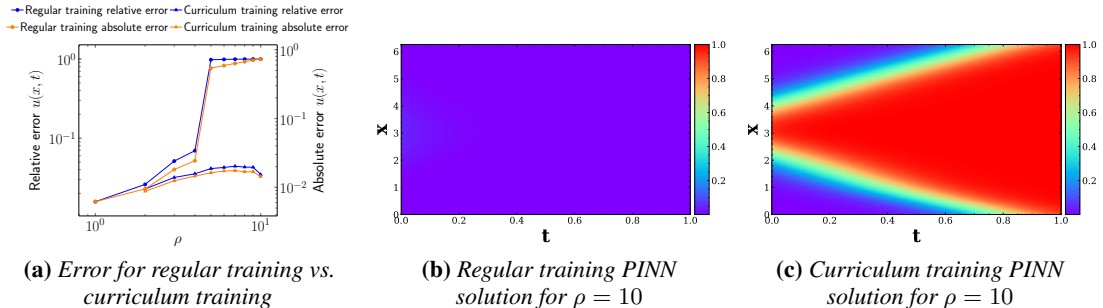

**(a)** *Error for regular training vs. curriculum training*  **(b)** *Regular training PINN solution for $\rho = 10$*  **(c)** *Curriculum training PINN solution for $\rho = 10$*

**Figure E.4:** *Curriculum regularization for 1D reaction from §A. The PINN is now able to predict the solution much more closely, including capturing the "sharp" features (traditionally hard for PINNs), and error is 1-2 orders of magnitude lower when using curriculum training over regular training.*

### E.3  Extra sequence-to-sequence learning results

We include sequence-to-sequence learning results for convection ( §3.1) in Tab. E.1.

|  |  | Entire state space | $\Delta t = 0.05$ | $\Delta t = 0.1$ |
|---|---|---|---|---|
| $\beta = 30$ | Relative error | $7.38 \times 10^{-1}$ | $2.13 \times 10^{-1}$ | $\mathbf{1.05 \times 10^{-1}}$ |
|  | Absolute error | $5.57 \times 10^{-1}$ | $1.29 \times 10^{-1}$ | $\mathbf{5.95 \times 10^{-2}}$ |
| $\beta = 40$ | Relative error | $8.25 \times 10^{-1}$ | $4.58 \times 10^{-1}$ | $\mathbf{2.41 \times 10^{-1}}$ |
|  | Absolute error | $6.06 \times 10^{-1}$ | $2.58 \times 10^{-1}$ | $\mathbf{1.35 \times 10^{-1}}$ |

**Table E.1:** *Predicting the entire state space versus discretizing the state space (i.e., seq2seq learning) for 1D convection ( §3.1). The seq2seq learning achieves lower error for both $\Delta t = 0.05$ and $\Delta t = 0.1$, in comparison to the PINN's approach of predicting the entire state space at once. For this example only, we use a higher number of collocation points (10000) for regular PINNs and seq2seq learning, which minimizes the variance in the seq2seq results.*

We include sequence-to-sequence learning results for reaction ( §A) in  Tab. E.2.

| | | Entire state space | $\Delta t = 0.05$ | $\Delta t = 0.1$ |
|---|---|---|---|---|
| $\rho = 5$ | Relative error | $9.79 \times 10^{-1}$ | $\mathbf{7.06 \times 10^{-2}}$ | $7.09 \times 10^{-2}$ |
| | Absolute error | $5.40 \times 10^{-1}$ | $2.52 \times 10^{-2}$ | $\mathbf{2.39 \times 10^{-2}}$ |
| $\rho = 6$ | Relative error | $9.88 \times 10^{-1}$ | $8.25 \times 10^{-2}$ | $7.78 \times 10^{-2}$ |
| | Absolute error | $5.88 \times 10^{-1}$ | $3.02 \times 10^{-2}$ | $\mathbf{2.65 \times 10^{-2}}$ |
| $\rho = 7$ | Relative error | $9.92 \times 10^{-1}$ | $8.16 \times 10^{-2}$ | $\mathbf{7.56 \times 10^{-2}}$ |
| | Absolute error | $6.31 \times 10^{-1}$ | $3.03 \times 10^{-2}$ | $\mathbf{2.69 \times 10^{-2}}$ |
| $\rho = 8$ | Relative error | $9.94 \times 10^{-1}$ | $8.19 \times 10^{-2}$ | $\mathbf{7.44 \times 10^{-2}}$ |
| | Absolute error | $6.69 \times 10^{-1}$ | $3.10 \times 10^{-2}$ | $\mathbf{2.73 \times 10^{-2}}$ |
| $\rho = 9$ | Relative error | $9.95 \times 10^{-1}$ | $\mathbf{7.02 \times 10^{-2}}$ | $8.63 \times 10^{-2}$ |
| | Absolute error | $7.02 \times 10^{-1}$ | $\mathbf{2.83 \times 10^{-2}}$ | $3.21 \times 10^{-2}$ |
| $\rho = 10$ | Relative error | $9.96 \times 10^{-1}$ | $\mathbf{6.88 \times 10^{-2}}$ | $7.47 \times 10^{-2}$ |
| | Absolute error | $7.31 \times 10^{-1}$ | $\mathbf{2.85 \times 10^{-2}}$ | $\mathbf{2.85 \times 10^{-2}}$ |

**Table E.2:** *Predicting the entire state space versus discretizing the state space (i.e., seq2seq learning) for 1D reaction ( §A). The seq2seq learning achieves lower error for both $\Delta t = 0.05$ and $\Delta t = 0.1$, in comparison to the PINN's approach of predicting the entire state space at once.*