# OpenReview forum: "Characterizing possible failure modes in physics-informed neural networks"
_NeurIPS.cc/2021/Conference — NeurIPS 2021 Poster_

### Official Review · Reviewer_sTZN · 2021-07-14

**Rating:** 7
**Confidence:** 3

**Summary:**

The paper studies the behavior of Physics Informed NNs (PINNs) on two common PDEs of physical relevance. It observes that PINNs fail to learn a good solution under standard training regime when the convection coefficient or viscosity coefficient is high, even when the closed-form solution does not depend on the constants.

The authors then seek to explain this behavior from an optimization standpoint. They plot the loss landscape along the two dominant hessian eigenvectors. They hypothesize that the optimization problem is ill conditioned due to the regularization term in the optimization objective that enforces the pde constraint.

They also run experiments to argue that sequence to sequence learning might work better for these ill conditioned problems.

**Limitations And Societal Impact:**

Limitations mentioned above.

**Main Review:**

I think the paper tackles an important problem in PINNs, but it could go deeper in its investigations.

- When diagnosing whether the loss landscape is ill conditioned, would it be possible to get more information about the spectrum of the hessian? It would be valuable to know bounds on the condition number, or some other investigations on the eigenvalue spectrum.
- The authors give a heuristic scaling law for the condition number. Could you give a bit more detail on the derivation? Would it be possible to empirically verify this? You could compute the condition number for the operator discretized on a finer and finer grid.
- The authors mention the condition number of the PDE operator, but isn't what we care about the condition number wrt the NN parameters? During optimization, we change the NN parameters and hope that doesn't lead to large changes in output. It would be interesting to know how the conditioning changes depending on NN architecture.
- In the warm start experiments, from my understanding you only run these on problems where the solution does not depend on the parameter. In that case, wouldn't the weights not change when you move to a larger coefficient problem, since you warm start from the solution? It would be interesting to see warm start experiments on problems where the solution does depend on the coefficients.
- The sequence to sequence solution at the end is mentioned, but the results seem mixed. It seems to work much better for diffusion vs convection. Do you have any insight on why this could be the case?

I am borderline in this paper, currently leaning towards a rejection, but open to changing my score if most of my comments above are addressed.

**Time Spent Reviewing:**

3 hours

---

> ### Author Response · Authors · 2021-08-10
> **Response to Reviewer sTZN**
>
> $>>>$ Question: would it be possible to get more information about the spectrum of the hessian?
>
> Answer: That is an excellent suggestion. We have computed the Hessian trace during training which
> shows a clear trend that as the coefficients ($\beta$ or $\nu$), and/or $\lambda$ value is increased,
> the spectrum becomes larger. For example, in the case of convection (from 3.1.1), for $\beta=0.1$ the Hessian trace of the converged model is $5 \times 10^{3}$, whereas for $\beta=30$ it is $1 \times 10^{5}$ (which is 1-2 orders of magnitude larger). We will add these results in the paper.
>
> As an additional update, we have included a figure here measuring the Hessian trace during training for the convection case (3.1.1) when  $\beta=20$ (a regime where the regular PINN performs poorly). As you can see, the Hessian trace gets progressively larger over training with regular PINNs. In contrast, the Hessian trace is an order of magnitude smaller with the curriculum learning approach.
>
> Figure link (Hessian spectrum): https://i.ibb.co/kBNTSwJ/Hessian-spectrum-beta20-curriculum.png
>
> $>>>$ Question: More detail on the derivation of condition number
>
> Answer: We have provided the derivation steps for the condition number in section E of the Appendix. We emphasize that those results by themselves are not new, and are known in the scientific computing
> area.
>
> $>>>$ Question: Isn't what we care about the condition number w.r.t. the NN parameters?
>
> Answer: Not necessarily. When we backpropagate through the loss, we are implicitly
> backpropagating through the PDE operator. If the operator (specifically its adjoint) is ill-conditioned,
> then there will be numerical stability which will propagate downstream to the NN parameters. This means that all the other gradients will become incorrect due to the numerical instability. It is also easy to see this intuitively. Consider a diffusion problem, where the PDE diffuses the information/state-variable through time. This PDE by nature is inherently ill-conditioned as the information is exponentially decayed/lost.
>
> $>>>$ Question: It would be interesting to know how the conditioning changes depending on NN architecture.
>
> Answer: That is an excellent point. There is certainly a dependence on the NN architecture. However,
> we did not investigate the role of the architecture as it adds a lot of additional moving parts  and requires significant additional
> results (and compute) which would not fit within one paper. Studying the NN architecture is part
> of the future work for our roadmap.
>
> $>>>$ Question: In the warm start experiments, from my understanding you only run these on problems where the solution does not depend on the parameter.
>
> Answer: You are right. We have tested the warm start (aka curriculum learning) on another problem (where the solution does depend on the parameter) and reported results in the official comment above (titled Extra Results, in Section 1). As those results clearly show, the warm start can significantly help including cases where the solution does change with the coefficients.
>
> $>>>$ Question:  Seq2Seq results seem mixed, and seem to work better for diffusion than convection. Why is that?
>
> Answer: This is somewhat expected since for the diffusion problem the information gets lost in time, and if we train on the entire state-time (as done in regular PINN), the model will have training data from distant
> points in time. While these points are indeed correlated, their mutual information is exponentially
> decayed due to the nature of diffusion operator. However, this is not the case for the convection problem, where the signal is propagated in time without diffusion (assuming no errors, of course).
>
> However, we also wanted to mention that the Seq2Seq still helps for the convection case.
> In the original paper we had tested with relatively small values of $\beta$, but we repeated those with higher convection coefficients and we indeed see an order of magnitude improvement as was reported in
> the official comment above (Section 3).

---

> > ### Comment · Reviewer_sTZN · 2021-08-19
> > **Response to response**
> >
> > Hi authors,
> >
> > Thank you for answering my questions. After looking at the other reviews, re-reading the paper, and looking at the extra results, I have changed my mind about the paper and am now in favor of acceptance.
> >
> > I think this paper would be a valuable contribution to the PINN literature. It identifies a significant problem, empirically investigates it, and proposes solutions that lead to demonstrable improvements. In particular, I am much more convinced about the seq2seq and curriculum learning cases after the additional results, and the loss landscape investigation.

---

### Official Review · Reviewer_4iUv · 2021-07-14

**Rating:** 7
**Confidence:** 4

**Summary:**

The authors provide a detailed empirical analysis of the numerical difficulties encountered when training so-called "physics informed neural networks" (PINNs), a recent class of models which incorporate physical (often differential) equations into the loss function of a neural network. The authors perform a systematic analysis on two relatively simple but common use-cases in scientific literature, convection and diffusion equations, and demonstrate the PINNs fail to adequately learn the underlying dynamics of the system even in simple cases. They show that these failures are due to the effect of the convection/diffusion coefficients and forcing terms on the conditioning of the dynamical system, i.e. larger coefficients or forcings tend to make the problem increasingly ill-conditioned, thereby impacting characteristics of the loss function. The authors then propose two possible workarounds for these issues; improved initialization via a warm start or preconditioning and re-framing the problem as a sequence-to-sequence learning task.

**Limitations And Societal Impact:**

Limitations and direction for future work are clearly discussed. I don't see any major societal impact issues, outside of perhaps concerns about potential misuse of PINNs in scientific research. However, the authors' work should, if anything, help ameliorate this.

**Main Review:**

Overall, the paper is very well written, the experiments are thorough and well designed, and the results are both interesting and of great practical value for researchers and practitioners in scientific machine learning. The theoretical analysis of the Lagrange optimization problem, while relatively informal, provides valuable insight to readers who are less familiar with the field of non-convex optimization and should encourage researchers to think carefully about how arbitrary regularization terms affect the underlying optimization problem in PINNs and other similar methods.

Ultimately, the results presented by the authors might not be terribly surprising to those with a background in scientific/numerical computing, where the problem of ill-conditioned differential operators in numerical integration and optimization has been grappled with for decades. However, it's important to confirm intuition and expectation with rigorous experimentation, which is exactly what the authors have done here.

In summary, this work provides valuable experimental confirmation and analysis of pathologies in PINNs and will, I think, be a valuable contribution to the field. The paper appears pretty much ready to publish as-is, but I will make a few suggestions below.

1. I missed a discussion on discretization strategies, both for the temporal and spatial derivatives. It is briefly mentioned in the supplementary materials that a uniform spatial grid spacing is used for convection (?), but that's all I can find. This is somewhat interesting since uniform grid spacing is typically the easiest case, so if the PINN can't even do that correctly, then it would be reasonable to expect it to fail on non-uniform grids as well. However, I would think that the resolution of the discretization would affect the numerical stability of the loss term, as finer meshes would result in more accurate gradients (assuming finite differencing is used) but also potentially more floating point error. Ideally, it would be nice to see an experiment demonstrating the sensitivity of the results presented here to discretization choices, but a brief discussion would also suffice, I think.


2. It would be nice to see a few more details on training, e.g. the number of training samples (collocation points?), discretization details (as mentioned previously), the convergence criterion, etc.


3. The results in figure D.1 are a bit puzzling to me. The authors claim "As the regularization parameter is increased, the loss landscape becomes increasingly more complex and harder to optimize (see the z-axis scale)" but this supposed increase in complexity is not apparent (at least to me) in the plots. The value of lambda that produced the lowest error values ($\lambda = 0.01$) has a very similar loss landscape plot to the worst value ($\lambda = 10.0$). In the latter case, the loss values are of course much higher, but that's to be expected as the value of $\lambda$ itself partially determines the scale of the loss term. It would be more informative to see a breakdown of how each loss term (initial condition, boundary conditions, and the unscaled differential equation) changes in response to the value of $\lambda$. I don't think the raw loss value is very informative.

4. Minor stylistic suggestion: The qualifier "Interestingly," is used, in my opnion, a bit too frequently throughout the text.

5. I was a bit confused to find empty sections in the appendix and had to go to the supplementary material to find the figures/tables. I'm not sure if this is a standard formatting choice of NeurIPS or not, but I would suggest at least adding notes directing the reader to the supplementary materials, or removing the section headers from the main document altogether.

**Time Spent Reviewing:**

4

---

> ### Author Response · Authors · 2021-08-10
> **Response to Reviewer 4iUv**
>
> $>>>$ Question: Details on training (the number of training samples (collocation points?), discretization details, the convergence criterion, etc.)
>
> Answer: We apologize for the confusion and we will include this in the final version. The grid that we used had 100 points in the t domain ($\Delta t =0.01)$, and 256 points in the x domain $(\Delta x = 0.004$). We found that for the ``failure modes'' we described, a finer resolution actually did not make a large difference. However, we agree that it would be useful to study convergence to the limit of an infinitely fine temporal and spatial grid. We will add clarifying information on the spatial and temporal grid that we chose to the manuscript.
>
> For our experiments, we used 1000 collocation points but found that using all the collocation points (around 25000) did not help reduce error very much for the failure modes we identified. We used the LBFGS optimizer with convergence criterion that included a tolerance gradient change of $10^{-7}$. We will add this clarifying information to the manuscript.
>
> $>>>$ Question: Uniform grid spacing is typically the easiest case, so if the PINN can't even do that correctly, it can fail on non-uniform grids as well. Important to test on finer grids to see impact of numerical stability.
>
> Answer: These are excellent observations. For this problem indeed the condition number scales as the mesh size is increased, which is the expected behavior as the reviewer mentioned. This is probably
> the reason why the regular PINN training does not improve with increasing the collocation points.
> We will add further results in the final version and will discuss this.
>
>
> $>>>$ Question: The results in figure D.1 are a bit puzzling to me...  the supposed increase in complexity is not apparent
>
> Answer: We will clarify these plots in the paper, but note that the z-axis scale is significantly higher for larger values of $\lambda$ (even if we account for the scale of the $\lambda$ values). We plan to add an additional example where the difference is more clear. We also plan to add Hessian spectrum results which clearly shows that the optimization problem becomes harder as the $\lambda$ value is increased (which is inline with the loss landscape plots).
>
> $>>>$ Question: Appendix sections
>
> Thanks for this comment and apologies for the confusion.
> We will reorganize the appendix accordingly.

---

### Official Review · Reviewer_LJn4 · 2021-07-15

**Rating:** 7
**Confidence:** 4

**Summary:**

In this paper the authors inspect the popular Physics Informed Neural Network (PINN) set of models and characterize their behavior (specifically their failure modes) in the context of two popular PDE systems, namely diffusion and convection. In each case, the authors demonstrate that PINNs learn effective representations in simple regimes while failing to learn representations of the PDE systems in more complex regimes. In addition, authors also demonstrate that it is not the inability of the neural network to fit the functions (due to some potential inability to represent non-trivial functions), but rather the way the PINNs incorporate the PDEs into the learning pipeline that is the reason for the failure. Finally, the authors also propose casting the PINN problem as a sequential prediction task (i.e., predicting all collocation points per time period) as opposed to tasking the neural network with simultaneously learning the full spatial and temporal domain.


**Limitations And Societal Impact:**

1. The proposed results although clearly demonstrative of certain failure modes in PINNs, offer an incomplete characterization and the solutions proposed (specifically the sequence-to-sequence approach) to alleviate the failure models is not fully detailed and demonstrated for sophisticated higher order PDEs (e.g., second order coupled PDEs like Navier-Stokes equation).

2. The claim that the reason for PINN based failure is due to the “subtle problems”  introduced through the PDE based soft constraint of L_F is not fully characterized or explained. What are the set of “subtle problems” that are being discussed? One that is definitely demonstrated is that the loss-landscape gets more complicated. However, other problems could have to do with the interplay between the loss terms which aren’t discussed.

    Speaking specifically to the point of the “interplay” between loss terms at different values of equation coefficients, it is imperative to demonstrate the influence of each of these loss terms on PINN learning dynamics to fully contextualize the failure. For example: at different values of $\beta$ (or $\nu$), how do the gradient distributions change w.r.t the gradient distributions of the other terms in the loss?

4. The model re-formulation in 3.1.2 and 3.2.2 is unclear. For example, in the case of 3.1.2, are the $L_B$ , $L_U$ terms replaced by those in equation 9 and the value of the forcing term `q` replaced with the new value on line `178`? If so, wouldn’t $\beta$ although absent in the analytical solution due to detailed simplifications), still have an influence on the PINN results?
Specifically (assuming the above point of $\beta$’s influence on the PINN gradient holds true), it should be inspected how the gradient dynamics with increasing $\beta$ (and correspondingly for increasing $\nu$ in section 3.2.2) influence the learning especially at the initial training epochs of the network. Have these dynamics been inspected and if they have, what are the learnings?

5. Has the addition of standard regularization terms (`L1, L2`, in conjunction with current PINN losses) been inspected and if so does that have a positive effect on the learned representation?


6. There has been some previous work inspecting the effect of various physics based regularizers in PINN [2] and certain adaptive learning rate schemes have been proposed and shown to be effective. It is important to contextualize the current set of results in the context of this previously published work to fully characterize the failure modes of PINNs.

**References:**

[1]  Raissi M, Perdikaris P, Karniadakis GE. Physics-informed neural networks: A deep learning framework for solving forward and inverse problems involving nonlinear partial differential equations. Journal of Computational Physics. 2019 Feb 1;378:686-707.

[2] Wang S, Teng Y, Perdikaris P. Understanding and mitigating gradient pathologies in physics-informed neural networks. arXiv preprint arXiv:2001.04536. 2020 Jan 13.



**Main Review:**

Overall the paper is clearly written and appropriately organized. But for a few minor typos, there are not many problems with the text or narrative structure.

**Positives:**
1. In this paper, authors tackle an important question of characterizing the performance of the popular PINN model [1], in the context of two canonical PDE systems, namely the convection and diffusion PDEs and do indeed demonstrate failure modes of PINNs.


2. Specifically, authors work in the context of 1D PDEs and showcase that for increasing values of equation coefficients (i. \beta i.e., convection coefficient in the case of the convection equation; ii. \nu i.e., viscosity in the case of the diffusion equation), PINN behavior degrades. Figures 1,3 in the paper demonstrate how the PINN prediction errors scale with increase in value of the equation coefficients (i.e., convection coefficient and viscosity respectively).



3. Further, the authors in their experimental setup section clearly outline a rigorous evaluation procedure wherein results have been calculated based on multiple runs of a PINN model optimized with L-BFGS and executed using different random seeds.


**Time Spent Reviewing:**

7 hours

---

> ### Author Response · Authors · 2021-08-10
> **Response to Reviewer LJn4**
>
> $>>>$ Question: The proposed methods may not work for more complex cases such as Navier Stokes?
>
> Answer: We kindly emphasize that we did not claim that our methods can solve any complex PDE such as Navier-Stokes, which is a highly difficult problem to solve with existing methods. Instead, we pointed out the problems in PINNs, and showed several cases where the seq2seq and/or curriculum learning can significantly outperform regular PINN training.
>
> Nevertheless, we have also added experiments for a more difficult task, a reaction-diffusion equation which has a non-linear term. We did find a significant improvement with Seq2Seq learning, as shown in the official comment (titled Extra Results) in Section 2.
>
> $>>>$ Question: It is imperative to demonstrate the influence of each of these loss terms on PINN learning dynamics to fully contextualize the failure
>
> Answer: This is definitely one of the subtle problems in PINNs which has been also suggested in Wang. et al [2], where an adaptive weighting strategy was suggested.
> While these results are certainly interesting, we did find other cases where adaptive weighting
> does not resolve the PINN training. For example, in the convection case (3.1.1), we found that for the high $\beta$ failure mode, the loss terms were all of similar orders of magnitude.
>
> $>>>$ Question: Doesn't the problem setup in Section 3.1.2 still depend on $\beta$ because it is in the residual loss term ($L_F$)?
>
> Answer: This is correct, while the solution to the state variable is independent of $\beta$, the different terms in the PDE do change as $\beta$ is varied.
>
> $>>>$ Question: Impact of other regularization terms?
>
> Answer: We did test different regularization schemes such as L1/L2 and they do not solve the PINN training problem.
>
> $>>>$ Question: It is important to contextualize the current set of results in the context of this previously published work to fully characterize the failure modes of PINNs including [2]
>
> Answer: We agree that [2] was interesting work, and we accordingly did cite the work of [2].
> In particular, the work of [2] shows that adaptive learning rate schemes have been shown to be effective for certain PINN problems. However, this approach does not always work (for example, it doesn't seem to help very much for the convection case failure mode with high $\beta$). We provided orthogonal solutions such as Seq2Seq and Curriculum learning which could be potentially combined/studied with adaptive loss weighting.

---

### Official Review · Reviewer_Cx4H · 2021-07-15

**Rating:** 6
**Confidence:** 4

**Summary:**

This work studies two specific (yet essential) PDE learning cases in the PINN framework. Notably, the work proposes two main conclusions to explain the difficult training in PINN networks:
- the loss landscape varies greatly with the PDE coefficient values (and works well with only low diffusion/conduction coefficient),
- the loss landscape (at the end of training) varies greatly with the physics informed penalty coefficient.

The main explanation to such difficulties lies in the condition number associated to PINN penalty. The work proposes two solutions: a warm-up, and a sequence to sequence approach.

**Limitations And Societal Impact:**

see above.

**Main Review:**

First, I highlight that the paper addresses a significant problem of sciML community.
The presented work is well-written and straightforward. The presented arguments are sound and can be leveraged by the community to address the evoked problems.

Concerning the proposed solutions. The investigation of Section 5.1 is interesting but only superficial, and therefore cannot lead to a solid conclusion. Also, the presentation of the solution 5.2 is in my opinion unclear.

Two main flows appears in the paper:
1) The causes of the flows in learning PINNs are well identified however it is difficult to state whether the evoked flows were taken into consideration when designing the solutions.
2) The part on the sequence-to-sequence learning is very poorly described and not clear, despite being a possible solution to the described problems even though experimental results are not compelling.


My questions are:
1) About the warm states: how come such warm states perform better ? What is the distribution of the weights at the end of training vs in the beginning or vs a regular initialization ?
Can the gradients of the parameters with respect to the loss be investigated in such a case ?
2) A simple approach to the problem would be to gradually increase the PINN penalty, progressively enforcing the constraint. Can such a simple approach solve the condition problem ?
3) Despite the vanishing/exploding gradients in RNN, the NLP community successfully leveraged simple optimization tricks such as gradient clipping ? Can the authors comment on the possible applicability of such optimization strategy ?
4) Can’t interior domain points for supervision help solving the ill-condition of the learning ?


____
### post rebuttal
____

The novel experimental results shows the advantage of the proposition to solve the highlighted issue.
I have decided to raise my score to 6.


**Time Spent Reviewing:**

7

---

> ### Author Response · Authors · 2021-08-10
> **Response to Reviewer Cx4H**
>
> $>>>$ Question: Seq2Seq results and/or curriculum learning results are not compelling.
>
> Answer: Please see the official comment (titled Extra Results) where we have provided additional results for Seq2Seq and curriculum learning.
>
> $>>>$ Question: Why warm start performs better?
>
> Answer: Warm start is a common method that has been found useful in other contexts, mostly
> due to the fact that we are dealing with a non-convex optimization problem.
> With warm starting, we start with lower values of $\beta$, $\nu$ which as we showed in the
> paper result in simpler loss landscapes (see Figure 5). One could also view these
> parameters as a temperature knob (as we move from lower values of $\beta$, $\nu$ to higher values), in which the warm start may have similarities with
> simulated annealing.
>
> $>>>$ Question: What is the distribution of the weights at the end of training vs in the beginning or vs a regular initialization?
>
> Answer: Our preliminary analysis does not show a clear trend in terms of the difference between the distribution of the weights, which tend to follow a normal distribution (also please note that we did not make any claim related to this in our paper).
>
> $>>>$ Question: Can the gradients of the parameters with respect to the loss be investigated in such a case ?
>
> Answer: This is an excellent question. The gradients do not necessarily become very large. In some of the cases, the issue is related more to getting stuck in high-order saddle points or getting stuck
> in local minimas.
>
> $>>>$ Question: Does gradually increasing the PINN penalty solve the problem?
>
> Answer: This is a very interesting suggestion. So far, we have found that this approach does not work
> out of the box, although it can be probably be useful in combining it with other approaches such
> as seq2seq and/or curriculum learning.
>
> $>>>$ Question:  Can gradient clipping solve the problems?
>
> Answer: As mentioned above, the problem is not necessarily due to exploding/vanishing gradients.
>
> $>>>$ Question: Can’t interior domain points for supervision help solving the ill-condition of the learning ?
>
> Answer: We are not sure what the reviewer is referring to as ''interior domain points." If the question is about increasing the number of ''collocation points," then we actually found that this cannot resolve the problems observed in the paper. This is somewhat expected as the condition number of the problem is not going to get better if we increase the collocation points where the PDE residual is to be computed.

---

> > ### Comment · Reviewer_Cx4H · 2021-08-12
> > **Reponse to the Authors**
> >
> > Post response update:
> >
> > ## Novel experimental Results
> > After a careful reading of the author's response and novel experiments, the proposed solution to solve PINN training seems rather efficient, most notably for _curriculum training_. The _seq2seq_ results are an order of magnitude better than PINN.
> >
> > Interestingly, the _seq2seq_ learning scheme is adaptable to cases where the space of the parameters is of higher dimension. However, sampling the parameters space in higher dimensions for _curriculum learning_ can be trickier. Nonetheless, the training scheme is still unclear to me, despite the explanation of the authors, i.e is the prediction at $t_1 = t_0 + \Delta t$ used as a novel initial condition for PINN training ?
> >
> > ## Theoretical insights
> > The major drawback of the paper in my opinion remains: the lack of link between the theoretical analysis illustrating the possible failures of PINN training and the proposed solutions.
> > Also I agree with Reviewer STZN that including more elements on the Hessian spectrum would provide valuable complementary information.
> >
> > While I understand that studying the _seq2seq_ setting can be difficult, complementary analysis in the case of _curriculum learning_ should be conducted in order to provide more insights to explain the relevance of this proposition.
> >
> > Therefore, despite interesting considerations, I chose to maintain my score.

---

> > > ### Author Response · Authors · 2021-08-13
> > > **Response to follow up reviewer comment (Hessian spectrum, loss landscapes included to show link between PINN failure modes and the success of curriculum learning)**
> > >
> > > First, we want to appreciate the reviewer's quick follow-up and for providing us their updated feedback.
> > > Below, we provide further details in light of this.
> > >
> > > $>>>$ Question: sampling the parameters space in higher dimensions for curriculum learning can be trickier.
> > >
> > > Answer: We are not sure why the problem should be harder, since in curriculum learning we are only changing the coefficients ($\beta$, or $\nu$). In our experiments, with different numbers of collocation points, we did not see any difference in the trends.
> > >
> > > $>>>$ Question:  The training scheme is still unclear to me, despite the explanation of the authors, i.e is the prediction at  t = t+Dt used as a novel initial condition for PINN training?
> > >
> > > This is correct. In Seq2Seq, the time horizon is split into time segments of size $\Delta t$ (for example, if T = [0, 1] and $\Delta t = 0.1$, we have 10 segments). Then, the prediction at $t= t + \Delta t$ is used as the initial condition for the next section.
> > >
> > > $>>>$ Question: Link between theoretical analysis of why PINNs fails and the proposed solution?
> > >
> > > Answer: This is an important and open question due to the many moving parts involved in training
> > > PINNs, which includes a combination of the NN architecture, its capacity, and the non-convex nature
> > > of the optimization problem which is made even more complex by the continuous regularization in PINNs.
> > >
> > > In our paper, we provided preliminary analysis on these factors, namely the connection between the
> > > ill-conditioned nature of the continuous regularization operator in PINNs and the optimization
> > > difficulty, which has been quantified both by the accuracy results, as well as by the new Hessian analysis (please see below).
> > > We also qualitatively showed how the loss landscape plots get affected, which is in line with the above observations. We think these results should help address your comment.
> > >
> > > $>>>$ Question: complementary analysis in the case of curriculum learning should be conducted in order to provide more insights to explain the relevance of this proposition... including more elements on the Hessian spectrum would provide valuable complementary information.
> > >
> > > Answer: We have further investigated the proposed curriculum learning through a loss landscape perspective to better understand why curriculum learning helps overcome some of the failures (and took into consideration the PINN flaws when designing this solution). We look at the convection example in 3.1.1. Here, we show the loss landscape for $\beta=30$ (a regime where the PINN does poorly) in the below figure, (a). It's clear that the optimization problem is very difficult and the optimizer ends up getting stuck in a flat regime.
> > >
> > > In contrast, we also show the loss landscape for $\beta=30$ ( (b) in the below figure) after it has been initialized with the weights via curriculum learning. As one can see, now the landscape is much smoother. This result provides further evidence that the optimization problem becomes relatively
> > > easier to solve with the curriculum learning approach.
> > >
> > > Figure link (landscapes): https://i.ibb.co/7QqwPk2/PINN-landscape-beta30-curriculum.png
> > >
> > > We have further quantified this by measuring the Hessian trace during training for the convection case (3.1.1) when $\beta=20$ (a regime where the regular PINN performs poorly) in the below figure.
> > > As you can see, the Hessian trace gets progressively larger over training with regular PINNs.
> > > In contrast, the Hessian trace is an order of magnitude smaller with the curriculum learning approach.
> > >
> > > Figure link (Hessian spectrum): https://i.ibb.co/kBNTSwJ/Hessian-spectrum-beta20-curriculum.png

---

> > > > ### Comment · Reviewer_Cx4H · 2021-08-19
> > > > **Response to complementary experiments**
> > > >
> > > > Thanks to the authors for providing these valuable novel experimental results.
> > > >
> > > > 1- _Curriculum_:  The new experiments for landscape analysis are valuable for your proposition. I highly recommend the inclusion of the loss landscape in the main paper if possible. This experiment enable the reader to understand the link between the various theoretical analysis and the proposition to solve the highlighted issues.
> > > > Moreover, it shows interesting properties of curriculum training to convexify the loss landscape.
> > > >
> > > > 2- _seq2seq_:
> > > > For the  _seq2seq_ training, I recommend a clearer description of the method in the main section.  Also, complementary analysis can be conducted to highlight the importance of the proposition:
> > > >
> > > > * at the beginning we are closer to the true initial condition, hence $\mathcal{L}_{u_0}$ may dominate in the overall loss function.
> > > >
> > > > * due to the "recurrent" nature of the training (from my understanding of what is actually done) that in fact amortizes the relative influence of the differential constraint.
> > > >
> > > > * Since it is a key cornerstone of your paper:  maybe a landscape analysis.
> > > >
> > > >
> > > > The novel experiments for the curriculum setting that links the theoretical analysis to experiments while showing practical interest provide convincing arguments.
> > > >
> > > > Despite I believe the _seq2seq_ setting is a little left out, the _curriculum_ setting is well designed and investigated, thus I have decided to raise my score to 6.

---

> > > > > ### Author Response · Authors · 2021-08-25
> > > > > **Response to reviewer follow-up about seq2seq**
> > > > >
> > > > > $>>>$ Question: I highly recommend the inclusion of the loss landscape in the main paper if possible. This experiment enable the reader to understand the link between the various theoretical analysis and the proposition to solve the highlighted issues.
> > > > >
> > > > > Answer: Thank you for your constructive feedback. We will include the results from the rebuttal (along with additional loss landscapes) in the manuscript/supplementary material for the final version.
> > > > >
> > > > > $ >>>$ Question: For the  seq2seq training, I recommend a clearer description of the method in the main section.
> > > > >
> > > > > Answer: Thank you for this feedback. We have prepared a figure to better explain the seq2seq learning and how it is different than regular PINN training. We will also add clarification in the paper.
> > > > >
> > > > > Figure link: https://i.ibb.co/wSJ90Nb/seqseq-pinns.jpg
> > > > >
> > > > > $>>>$ Question: Also, complementary analysis can be conducted to highlight the importance of the proposition: At the beginning we are closer to the true initial condition, hence $\mathcal{L}_{u_0}$ may dominate in the overall loss function. Due to the "recurrent" nature of the training (from my understanding of what is actually done) that in fact amortizes the relative influence of the differential constraint.
> > > > >
> > > > > Answer: That is an excellent observation. It is true that in the beginning we are closer to the true initial condition, and at least for the first
> > > > > few segments, the impact of the collocation points, and their possible ill-conditioned impact, is less pronounced as compared to considering the full state-space.
> > > > > This could point to another potential reason why seq2seq is outperforming. By considering a shorter timespan, the NN needs to approximate a much simpler function than when it is learning the function to represent the full state-space. Therefore, by restricting the time horizon to only a $\Delta t$ amount, the problem becomes easier to solve (with the trade-off that there is a smaller number of collocation points). As such, we expect a trade-off of how small the time segments can be, which is exactly what we observe in our results.
> > > > >
> > > > > We should also note that this behavior has corollaries in scientific computing, where time marching schemes often outperform solving the entire state-space at once.

---

### Author Response · Authors · 2021-08-10
**Extra Results**

To Reviewers 1, 2, 3, 4:

We would like to thank all the reviewers for taking the time and providing their valuable feedback. Several reviewers seemed to have similar questions about Section 5 (curriculum learning and Seq2Seq),
and we address those shared questions here (individual responses to specific reviewer questions are replied to separately).

$>>>$ Reviewers Question: Curriculum learning results don't seem to give solid conclusions that it works.

Answer: In the interest of space, we included minimum results but we did also test additional cases which we report here and will report in the final version of the manuscript. These additional cases clearly show that curriculum learning provides improvement over regular PINN training. They are outlined below:

% -------------------

Section 1) 1D Convection Results (Curriculum Learning):

- $\beta=20, q=0$:

Regular PINN: Relative error $7.5\times 10^{-1}$, Absolute error $4.3\times 10^{-1}$

Curriculum learning: Relative error $9.8 \times 10^{-3}$, Absolute error $5.4 \times 10^{-3}$

- $\beta=30, q=0$:

Regular PINN: Relative error $9.0\times 10^{-1}$, Absolute error $5.4\times 10^{-1}$

Curriculum learning: Relative error $2.0 \times 10^{-2}$, Absolute error $1.0 \times 10^{-2}$

- $\beta=40, q=0$:

Regular PINN: Relative error $9.6\times 10^{-1}$, Absolute error $5.8\times 10^{-1}$

Curriculum learning: Relative error $5.3 \times 10^{-2}$, Absolute error $2.7 \times 10^{-2}$

Experimental Details: 1D convection results correspond to section 3.1.1. Curriculum learning is
performed by solving the problem with low values of $\beta$ and using the trained weights to initialize
training for higher values of $\beta$.

Conclusion:
We can clearly see that regular PINN training does poorly whereas the warm starting/curriculum learning approach actually decreases error by 1-2 orders of magnitude.

% -------------------

$>>>$ Reviewers Question: Seq2Seq results are not convincing.

Answer: We tested Seq2Seq on more complex settings, including reaction-diffusion systems, where Seq2Seq provides significant improvement over regular PINN training. These are outlined below:

% -------------------

Section 2) 1D Reaction-diffusion results (Seq2Seq):

- $\nu=0.0001$, $\rho=5$, $q=0$:

Regular PINN: Relative error $9.8 \times 10^{-1}$, Absolute error $5.4 \times 10^{-1}$

Seq2Seq, $\Delta t=0.1$: Relative error $6.8 \times 10^{-2}$, Absolute error $2.3 \times 10^{-2}$

- $\nu=0.0001$, $\rho=6$, $q=0$:

Regular PINN: Relative error $9.9 \times 10^{-1}$, Absolute error $5.9 \times 10^{-1}$

Seq2Seq, $\Delta t=0.1$: Relative error $8.0 \times 10^{-2}$, Absolute error $2.6 \times 10^{-2}$

- $\nu=0.0001$, $\rho=7$, $q=0$:

Regular PINN: Relative error $9.9 \times 10^{-1}$, Absolute error $6.3 \times 10^{-1}$

Seq2Seq, $\Delta t=0.1$: Relative error $8.5 \times 10^{-2}$, Absolute error $2.8 \times 10^{-2}$

- $\nu=0.0001$, $\rho=8$, $q=0$:

Regular PINN: Relative error $9.9 \times 10^{-1}$, Absolute error $6.7 \times 10^{-1}$

Seq2Seq, $\Delta t=0.1$: Relative error $8.7 \times 10^{-2}$, Absolute error $2.8 \times 10^{-2}$

- $\nu=0.0001$, $\rho=9$, $q=0$:

Regular PINN: Relative error $1.0 \times 10^{0}$, Absolute error $7 \times 10^{-1}$

Seq2Seq, $\Delta t=0.1$: Relative error $7.9 \times 10^{-2}$, Absolute error $2.8 \times 10^{-2}$

- $\nu=0.0001$, $\rho=10$, $q=0$:

Regular PINN: Relative error $1.0 \times 10^{0}$, Absolute error $7.3 \times 10^{-1}$

Seq2Seq, $\Delta t=0.1$: Relative error $7.0 \times 10^{-2}$, Absolute error $2.6 \times 10^{-2}$

Experimental Details: Given some $\Delta t$, we predict everything from a starting $t$ to the $\Delta t$. For example, if $\Delta t=0.1$, we predict everything from $t=0$ to $t=0.1$. We then use the predicted solution at $\Delta t = 0.1$ to predict from $t = 0.1$ to $=0.2$, and so on until we cover the whole domain.

For reaction-diffusion systems, the PDE here is $\frac{du}{dt} = \nu \frac{\partial^2 u}{\partial x^2} + \rho u - \rho u^2$.
The system has periodic boundary conditions, $u(0, t) = u(2\pi, t)$, and the initial condition is a Gaussian, $exp(- (x - \pi))^2/2(\pi/4)^2$. We looked at cases where $\nu=0.0001$ and $\rho = 5$ to $\rho = 10$, which is a regime where the regular PINN model has a relative error of 100\% and is unable to predict the solution at all (despite $\nu$ being so low).

Conclusions: We see that while regular PINN training does poorly, the seq2seq method decreases error by 1-2 orders of magnitude for reaction-diffusion systems. This includes for higher values of $\rho$, where the regular PINN is not able to capture the "sharp" transitions.

% -------------------

% -------------------

Section 3) 1D convection Results (Seq2Seq):

- $\beta=30, q=0$:

Regular PINN: Relative error $8\times 10^{-1}$, Absolute error $5.6\times 10^{-1}$

Seq2Seq, $\Delta t =0.1$: Relative error $1.0\times 10^{-1}$, Absolute error $5.9\times 10^{-2}$

- $\beta=40, q=0$:

Regular PINN: Relative error $9\times 10^{-1}$, Absolute error $6.1\times 10^{-1}$

Seq2Seq, $\Delta t = 0.1$: Relative error $2.4\times 10^{-1}$, Absolute error $1.3\times 10^{-1}$

Experimental Details: Same as above, but with convection (from 3.1.1) instead of reaction-diffusion. We also use more collocation points for this experiment to minimize the variance in the seq2seq results (the same number of collocation points are still used for both regular PINNs and seq2seq).

Conclusions: We see that while regular PINN training does poorly, the seq2seq method also decreases error for the convection examples.

% -------------------

$>>>$ Question:  Does the proposed curriculum and/or Seq2Seq learning method still work for *non-linear PDEs*?

Answer:  As the above results with reaction-diffusion show, we do see 1-2 orders of magnitude improvement
even for non-linear PDEs. However, we do emphasize that we are not claiming that our approach
may work on all kinds of PDEs, but that we show several use cases where we can
significantly improve over regular PINNs.

-----------------------------------------------------------------------------------------------------------------------------

$>>>$ Additional results (Hessian spectrum, landscape analysis) in response to reviewer questions: In the paper, we provided preliminary analysis for the connection between the ill-conditioned nature of the continuous regularization operator in PINNs and the optimization difficulty. We include additional results here exploring the link between the PINN failures and why curriculum learning helps overcome some of the failures, through a loss landscape perspective and through Hessian analysis. We look at the convection example in 3.1.1.

Here, we show the loss landscape for  $\beta=30$ (a regime where the PINN does poorly) in the below figure, (a). It's clear that the optimization problem is very difficult and the optimizer ends up getting stuck in a flat regime.

In contrast, we also show the loss landscape for $\beta=30$ ( (b) in the below figure) after it has been initialized with the weights via curriculum learning. As one can see, now the landscape is much smoother. This result provides further evidence that the optimization problem becomes relatively easier to solve with the curriculum learning approach.

Figure link (landscapes): https://i.ibb.co/7QqwPk2/PINN-landscape-beta30-curriculum.png

We further quantified this by measuring the Hessian trace during training for the convection case (3.1.1) when $\beta=20$ (a regime where the regular PINN performs poorly) in the below figure. The Hessian trace gets progressively larger over training with regular PINNs. In contrast, the Hessian trace is an order of magnitude smaller with the curriculum learning approach.

Figure link (Hessian spectrum): https://i.ibb.co/kBNTSwJ/Hessian-spectrum-beta20-curriculum.png

---

### Decision · Program_Chairs · 2021-09-27

**Decision:**

Accept (Poster)

**Comment:**

This paper addresses an interesting and important topic, and performs a careful and well motivated analysis, producing very practical lessons. There was active discussion of this paper -- 3 out of 4 reviewers raised their scores as a result, and all 4 reviewers vote for acceptance. I am therefore recommending that this paper be accepted.